# NEURAL NETWORKS BEHAVE AS HASH ENCODERS: AN EMPIRICAL STUDY

## ABSTRACT

The input space of a neural network with ReLU-like activations is partitioned into multiple linear regions, each corresponding to a specific activation pattern of the included ReLU-like activations. We demonstrate that this partition exhibits the following encoding properties across a variety of deep learning models: (1) *determinism*: almost every linear region contains at most one training example. We can therefore represent almost every training example by a unique activation pattern, which is parameterized by a *neural code*; and (2) *categorization*: according to the neural code, simple algorithms, such as $K$-Means, $K$-NN, and logistic regression, can achieve fairly good performance on both training and test data. These encoding properties surprisingly suggest that *normal neural networks well-trained for classification behave as hash encoders without any extra efforts.* In addition, the encoding properties exhibit variability in different scenarios. Further experiments demonstrate that *model size*, *training time*, *training sample size*, *regularization*, and *label noise* contribute in shaping the encoding properties, while the impacts of the first three are dominant. We then define an *activation hash phase chart* to represent the space expanded by model size, training time, training sample size, and the encoding properties, which is divided into three canonical regions: *under-expressive regime*, *critically-expressive regime*, and *sufficiently-expressive regime*.

## 1 INTRODUCTION

Recent studies have highlighted that the input space of a rectified linear unit (ReLU) network is partitioned into linear regions by the nonlinearities in the activations (Pascanu et al., 2013; Montufar et al., 2014; Raghu et al., 2017), where ReLU networks refer to the networks with only ReLU-like (two-piece linear) activation functions (Glorot et al., 2011; Maas et al., 2013; He et al., 2015; Arjovsky et al., 2016). Specifically, the mapping induced by a ReLU network is linear with respect to the input data within linear regions and nonlinear and non-smooth in the boundaries between linear regions. Intuitively, the interiors of linear regions correspond to the linear parts of the ReLU activations and thus corresponds to a specific activation pattern of the ReLU-like activations, while the boundaries are induced by the turning points. Therefore, every example can be represented by the corresponding activation pattern of the linear region where it falls in. In this paper, we parameterize the activation pattern as a 0-1 matrix, which is termed as *neural code*. Correspondingly, a neural network induces an *activation mapping* from every input example to its neural code. For the detailed definition of neural code, please refer to Section 3. This linear region partition still holds if the neural network contains smooth activations (such as sigmoid activations and tanh activations) besides ReLU-like activations, in which the interiors are no longer linear but still smooth.

Through a comprehensive empirical study, this paper shows:

> *A well-trained normal neural network performs a hash encoder without any extra effort, where the neural code is the hash code and the activation mapping is the hash function.*

Specifically, our experiments demonstrate that the neural code exhibits the following encoding properties shared by hash code (Knuth, 1998) in most common scenarios of deep learning for classification tasks:

- **Determinism:** When a neural network has been well trained, the overwhelming majority of the linear regions contain at most one training example per region. Thus, almost every training example can be represented by a unique neural code. To evaluate this determinism property quantitatively, we propose a new term *redundancy ratio*, which is defined to be $\frac{n-m}{n}$ where $n$ is the sample size and $m$ is the number of the linear regions containing the sample. Experimental results show that the redundancy ratio is near zero in almost every scenario.

- **Categorization:** The neural codes of examples from the same category are close to each other in the neural code space under the distance whereon (such as Euclidean distance and Hamming distance), while the neural codes are far away from each other if the corresponding examples are from different categories. We conduct clustering and classification experiments on the neural code space. Empirical results suggest that simple algorithms, such as $K$-Means (Lloyd, 1982), $K$-NN (Cover & Hart, 1967; Duda et al., 1973), and logistic regression can achieve fairly good training and test performance which is at least comparable with the performance of the corresponding neural networks on the raw data.

The two encoding properties collectively measure the expressivity of the activation mapping. For the brevity, we term this expressivity as *goodness-of-hash*.

It is worth noting that our study is different to the efforts of employing neural networks to learn hash functions, where the outputs are the hash codes of the input examples (Wang et al., 2017). Specifically, Xia et al. (2014); Lai et al. (2015); Zhu et al. (2016); Cao et al. (2017; 2018) design hash layers to neural networks for learning hash functions of images; and Simonyan & Zisserman (2014); Donahue et al. (2015); Wang et al. (2016); Varol et al. (2017); Chao et al. (2018); Yuan et al. (2019) extend the applicable domain to video data. Surprisingly, this paper reports that the activation pattern (or neural code) is already fairly good hash code.

The encoding properties also exhibit some variabilities in different scenarios. We then conduct comprehensive experiments to investigate which factors would influence the encoding properties. The empirical results suggest that *model size*, *training time*, *training sample size*, *regularization*, and *label noise* contribute in shaping the encoding properties, while the first three have dominant influences. Specifically, larger model size, longer training time, and more training data lead to stronger encoding properties.

We evertually define an *activation hash phase chart* to characterize the space expanded by model size, training time, sample size, and the goodness-of-hash. According to the discovered correlations, this space is partitioned into three canonical regions:

- **Under-expressive regime.** The redundancy ratio is considerably higher than zero while the categorization accuracy is considerably lower than $100\%$. However, both redundancy ratio and categorization accuracy exhibit significantly *positive* correlations with model size, training time, and training sample size.

- **Critically-expressive regime.** This is a transition region between the under-expressive and sufficiently-expressive regimes. The goodness-of-hash changes considerably as model size, training time, and sample size change, while the correlations become insignificant.

- **Sufficiently-expressive regime.** The redundancy ratio is almost zero while the categorization accuracy has become fairly good. One can hardly observe them change when model size, training time, and training sample size change. This regime covers many popular scenarios in the current practice of deep learning, especially those in classification.

It is worth noting that our partition is different from the one proposed by Nakkiran et al. (2020), which characterizes the the expressivity (or expressive power) of the input-output mapping induced by a neural network. By contrast, our the partition in activation hash phase chart characerizes goodness-of-hash.

Our results are established on empirical results of multi-layer perceptrons (MLPs), VGGs (Simonyan & Zisserman, 2015), ResNets (He et al., 2016a;b), ResNeXt (Xie et al., 2017), and DenseNet (Huang et al., 2017) trained for classification on the datasets MNIST (LeCun et al., 1998) and CIFAR-10 (Krizhevsky & Hinton, 2009). Our code is available in the supplementary material. The code, obtained models, and collected data will be released publicly.

## 2 RELATED WORKS

Many works have also studied the number of linear regions (linear region counting) in neural networks containing ReLU activations. Pascanu et al. (2013); Montufar et al. (2014) propose an upper bound exponential with the network's depth and polynomial to the width. Montufar et al. (2014); Arora et al. (2016); Hu & Zhang (2018); Hanin & Rolnick (2019b); Zhu et al. (2020) improve the upper bounds and lower bounds for the linear region counting. Xiong et al. (2020) study the linear region counting of convolutional neural networks. Serra et al. (2018) theoretically show that one can obtain a larger linear region counting when the layer monotonously decreasing from the early layer to the final layer. Poole et al. (2016); Novak et al. (2018); Hanin & Rolnick (2019a) investigate how the linear region counting would change as the training progresses. Raghu et al. (2017) define a trajectory length based on the activation patterns to measure the expressive powers of neural networks. Kumar et al. (2019) empirically demonstrate that a large proportion of the ReLU activations are always either activated or de-activated for all training examples in a well-trained and fixed network. Zhang & Wu (2020) report that optimization methods would also significantly influence the geometry property of linear regions.

A partition in the loss surfaces of neural networks has also been observed. Soudry & Hoffer (2018) highlighted that the loss surfaces of neural networks with piecewise linear functions are partitioned into multiple smooth and multilinear open cells, while the boundaries are non-differentiable. He et al. (2020) discovered three other properties: (1) every local minimum in a cell is the global minimum in the cell; (2) local minima in a cell are interconnected; and (3) all local minima in a cell are in an equivalence class. This paper focuses on another partition observed in the data space.

## 3 PRELIMINARIES

Suppose a ReLU network $\mathcal{N}$ is trained to fit a dataset $S = \{(x_i, y_i), i = 1, \ldots, n\}$ for classification, where $x_i \in \mathcal{X} \subset \mathbb{R}^{d_X}$, $d_X$ is the dimension of $x$, $y_i \in \mathcal{Y} = \{1, \ldots, d\}$, $d$ is the number of potential categories, and $n$ is the training sample size. Additionally, we assume that all examples $(x_i, y_i)$ are independent and identically distributed (i.i.d.) random variables drawn from a data distribution $\mathcal{D}$. Moreover, we denote the well-trained model as $\mathcal{M}$. Here, "well-trained" refers to the training procedure has converged.

Recent works have shown that the input space of a ReLU network $\mathcal{N}$ is partitioned into multiple linear regions, each of which corresponds to a specific activation pattern of the ReLU activation functions. In this paper, we represent the activation pattern as a matrix $\boldsymbol{P} \in \mathcal{P} \subset \{0, 1\}^{l \times w}$, where $l$ and $w$ are the depth and the largest width of this neural network $\mathcal{N}$, respectively. Specifically, the $(i, j)$-th component characterizes the activation statue of the $j$-th ReLU neuron in the $i$-th layer. The $(i, j)$-th component equals 1 represents that this neuron is activated, while it equals 0 means this neuron is deactivated or invalid[1]. The matrix $\boldsymbol{P}$ is termed as *neural code*. We can also re-formulate the neural code as a vector if no confusion of the depth and width occurs.

It is worth noting that the volume of boundaries between linear regions is zero, because the boundaries correspond to at least one turning point in the activations, which are of measure zero. Correspondingly, the probability that some examples fall in the boundaries is zero. We thus assume no example is in the boundaries. Therefore, fixing the weight $w$ of the model $\mathcal{M}$, every example $x \in \mathcal{X}$ can be indexed by the neural code $\boldsymbol{P}$ of the corresponding linear region. It is worth noting that the instance $x$ can be either seen in the training sample set or the test sample set.

## 4 NEURAL NETWORKS PERFORM AS HASH ENCODERS

Through an empirical study, this paper discovers that (1) the neural code is a hash code of the corresponding datum; and (2) correspondingly, the mapping $\mathcal{X} \to \mathcal{P}$ from datums to their neural codes is a hash function, which is termed as *activation mapping*. In contrast to the learning-to-hash methods, we find well-trained normal neural networks for normal tasks (such as classification) already perform hash encoding without any extra efforts. Specifically, the neural code exhibits

---

[1]Different layers may have different numbers of neurons. Therefore, there might be some indices $(i, j)$ are invalid. We represent the activation patterns of these neurons as 0 since they are never activated.

two major encoding properties: (1) *determinism*, and (2) *categorization*; please see more details in Section 1. Similar to the works in learning-to-hash, we adopt the following two measures to quantitatively evaluate the activation mapping as a hash mapping.

**Redundancy ratio.** We define the following *redundancy ratio* to measure the determinism property. Generally, a smaller redundancy ratio is preferred when we evaluating the activation mapping as a hashing function.

**Definition 1.** *(redundancy ratio) Suppose there are $n$ examples in a dataset $S$. If they are located in $m$ activation regions, the redundancy ratio is defined to be $\frac{n-m}{n}$.*

**Categorization accuracy.** Hash code is usually employed for nearest neighbor searching. In this paper, we perform simple algorithms, such as $K$-Means, $K$-NN, and logistic regression, to the neural codes. The training accuracy and test accuracy are employed to evaluate the encoding properties. Specifically, a higher accuracy corresponds to a better encoding property. It is worth noting that $K$-Means is designed for unsupervised learning. Here, we use it to verify our encoding properties in the context of supervised learning. The pipeline is modified and given in Appendix A.

We investigate the activation mappings induced by MLPs, VGG-18, ResNet-18, ResNet-34, ResNeXt-26, DenseNet-28 trained on the MNIST dataset and VGG-19, ResNet-18, ResNet-20, and ResNet-32 trained on the CIFAR-10 dataset. Details of the implementations are given in Appendix A due to the space limitation. The redundancy ratio is almost zero and the categorization accuracy is fairly good in all cases, as presented in Tables 1 and 2. These results verify the determinism and categorization properties. Moreover, we visualize the neural codes employing t-SNE (Maaten & Hinton, 2008), as presented in Figure 1. This visualization suggests that examples from the same category are concentrated together while clear boundaries are observed between examples from different categories, which coincide with the categorization property.

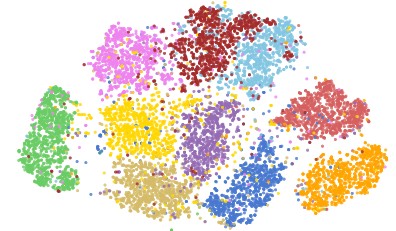

Figure 1: t-SNE visualization of the neural codes of MNIST generated by a one-hidden-layer MLP with width 100.

Table 1: Accuracy of $K$-Means and $K$-NN on the neural code of CNNs trained on MNIST.

| Architecture | $K$-Means acc | $K$-NN acc |
|---|---|---|
| VGG-18 | 99.95% | 99.33% |
| ResNet-18 | 98.96% | 99.32% |
| ResNet-34 | 99.66% | 99.49% |
| ResNeXt-26 | 98.31% | 99.24% |
| DenseNet-28 | 69.87% | 98.59% |

Table 2: Accuracy of logistic regression (LR) on the neural code of CNNs trained on CIFAR-10.

| Architecture | LR acc | Test acc |
|---|---|---|
| VGG-19 | 92.19% | 91.43% |
| ResNet-18 | 89.55% | 90.42% |
| ResNet-20 | 88.76% | 90.44% |
| ResNet-32 | 89.05% | 90.45% |

## 5 FACTORS THAT SHAPE OF ENCODING PROPERTIES

The results presented in Tables 1 and 2 also suggest that the encoding properties exhibit variability in different scenarios. Through comprehensive experiments, we investigate which factors would influence the encoding properties. The investigated factors include model size, training time, sample size, three popular regularizers, random data, and noisy labels. Some details of the experiment implementations are given in Appendix A due to the space limitation.

### 5.1 RELATIONSHIP BETWEEN MODEL SIZE AND ENCODING PROPERTIES

We first study how model size would influence the encoding properties. We trained 115 one-hidden-layer MLPs on the MNIST dataset and 200 five-hidden layer MLPs on the CIFAR-10 dataset with different widths (please see the full list of widths involved in our experiments in Appendix A), while all irrelative variables are strictly controlled. The experiments are repeated for 5 trials on MNIST and 10 trials on CIFAR-10, respectively.

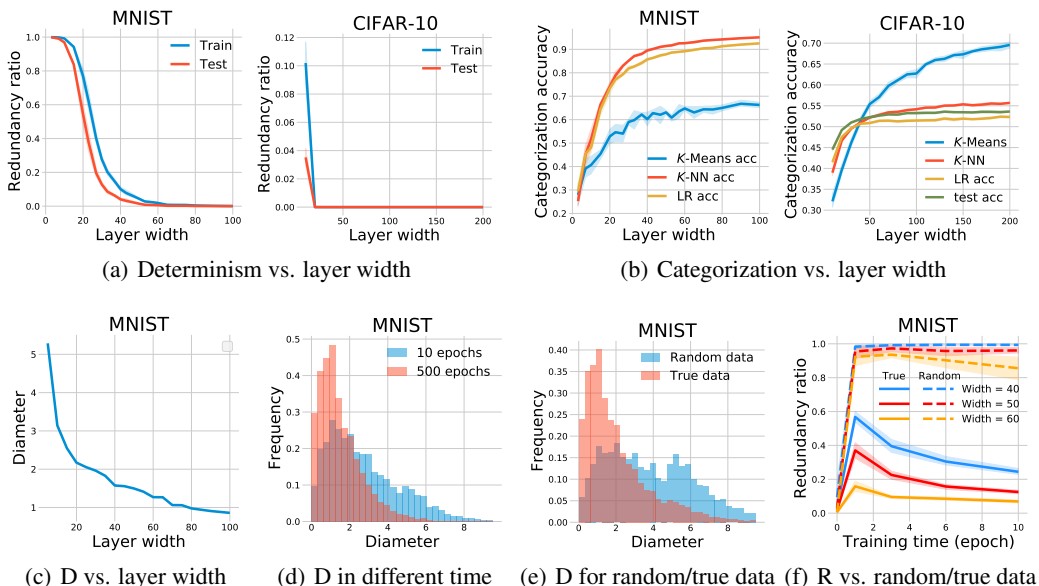

(a) Determinism vs. layer width    (b) Categorization vs. layer width

(c) D vs. layer width    (d) D in different time    (e) D for random/true data    (f) R vs. random/true data

Figure 2: (a) Plots of redundancy ratio as a function of the layer width of MLPs on both training set (blue) and test set (red). (b) Plots of test accuracies of $K$-Means (blue), $K$-NN (red), and logistic regression (LR, range) as functions of the layer widths of MLPs. (c) Plots of the average stochastic activation diameter (D) as a function of the layer width of MLPs on MNIST. (d) Histograms of stochastic diameters (D) calculated on MNIST for an MLP of width $50$ trained on MNIST for $10$ epochs (blue) and $500$ epochs (red), respectively. (e) Histograms of stochastic diameters (D) calculated on MNIST (blue) and randomly generated data with the same dimension (red), respectively, for an MLP of width $50$ trained on MNIST. The two red histograms are identical. (f) Plots of redundancy ratio (R) calculated on MNIST ("True data", solid lines) and randomly generated data (dotted lines) as functions of training time for MLPs of widths $40$ (blue), $50$ (red), and $60$ (orange). The dotted lines show networks trained on unaltered data, evaluated with random data. The models are trained for $5$ times on MNIST and $10$ times on CIFAR-10 with different random seeds. The darker lines show the average over seeds and the shaded area shows the standard deviations.

**Measure model size by width.** In the context of MLPs, a natural measure for the model size is the layer width.[2] We then calculate the redundancy ratio and categorization accuracy in all cases, as presented in Figures 2(a) and 2(b). From the plots, we can observe clear correlations between the encoding properties and the width: (1) the redundancy ratio starts at a relatively high position (nearly $1$ on both training and test sets of MNIST, around $0.1$ on the training set of CIFAR-10, and around $0.04$ on the test set of CIFAR-10). Then, it decreases to almost $0$ in all cases as the layer width increases; and (2) the categorization accuracy starts at a relatively low position (about $25\%$ on MNIST, and $32$-$45\%$ on CIFAR-10). Then, as the width increases, the accuracy monotonically increase in all cases to a relatively high position (around $70\%$ for $K$-Means on both datasets, higher than $90\%$ for $K$-NN and logistic regression on MNIST, and around $50\%$ for $K$-NN and logistic regression on CIFAR-10, similar to the test accuracy on the raw data).

**Measure model capacity by the diameters of linear regions.** We design an *average stochastic activation diameter* as a new measure for evaluating the model capacity, which is calculated in three steps: (1) we sample a random direction via the uniform distribution; (2) we define the *stochastic diameter* of a linear region to be the length of the longest intersected line segment of the linear region and the line along the sampled direction; and (3) we define the average stochastic activation diameter to be the mean of the stochastic diameters of all the linear regions containing data. Intuitively, a smaller average stochastic activation diameter means that the input space has been divided into

---

[2]Depth is also a natural measure for model size. However, the optimal training protocol (especially training time) for networks of different depths significantly differs. Thus, it is hard to conduct experiments on depth while controlling other factors.

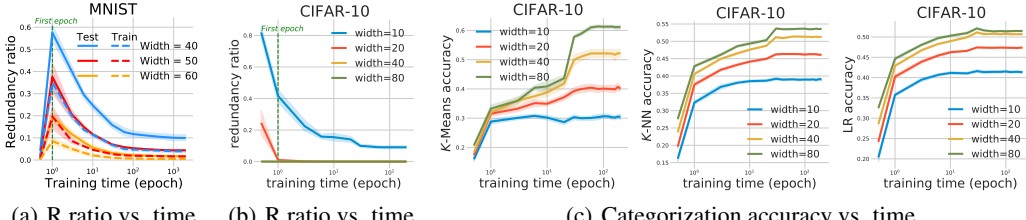

(a) R ratio vs. time     (b) R ratio vs. time     (c) Categorization accuracy vs. time

Figure 3: (a) Redundancy ratio (R ratio) as a function of training time on MNIST. (b) Redundancy ratio (R ratio) as a function of training time on CIFAR-10. (c) Test accuracy of $K$-Means (left), $K$-NN (middle), and logistic regression (right) as functions of training time. The models are MLPs of widths $10$ (blue), $20$ (red), $40$ (orange), and $80$ (green). The dotted lines show networks trained on unaltered data, evaluated with random data. The darker lines show the average over seeds and the shaded area shows the standard deviations.

smaller linear regions, and thus can represent more sophisticated data structures. Therefore, it can serve as a measure of model capacity. Correspondingly, a negative correlation between the layer width and the average stochastic activation diameter is observed, as illustrated in Figure 2(c).

Hanin & Rolnick (2019b) also define a 'the typical distance from a random input to the boundary of its linear region.' In contrast, our diameter is intuitively the longest distance between two points in a linear region. When the linear region is an ideal ball, their distance is equal to or smaller than the radius of the ball, the half of our diameter. However, linear regions are usually extremely irregular in practice. Please refer to a visualization of the linear regions in Figure 1, Hanin & Rolnick (2019b). Given this, the distances of Hanin & Rolnick (2019b) would be significantly smaller than our diameter. Overall, these two definitions would exhibit a significant discrepancy depending on the irregular level; one can be even fixed when the other is significantly changed. Moreover, their distance can yield a lower bound for the linear region volume, while ours can deliver an upper bound.

We also studied the encoding properties beyond the data generating distribution. A set of examples is generated according to the uniform distribution over the unit ball centered at the original point. The original data is also normalized so that every pixel is in the range $[0, 1]$. Therefore, the scales of the random data and the original are comparable. We observe that the redundancy ratio is larger than $0.8$ on the randomly generated data; see Figure 2(f). This result suggests that the determinism property no longer stands, and correspondingly, one cannot represent randomly generated data by unique neural codes. Therefore, the categorization property also becomes elusive. We further propose the following hypothesis to explain our findings.

**Hypothesis 1.** *The diameters of linear regions in the support of data distribution becomes smaller as the training progresses, while the diameters of regions far away do not change much.*

We then collect the average stochastic activation diameters for each scenario, as illustrated in Figure 2(d) and Figure 2(e). We observe that the stochastic diameters are more concentrated when the training time is longer; see Figure 2(d). Moreover, we observe an interesting result that the stochastic diameters for true data is more concentrated at lower values than the stochastic diameters for random data. Figure 2(e) shows a diagram for stochastic diameters. The diagrams for other scenarios are given in Appendix B. These results fully support our hypothesis.

## 5.2 RELATIONSHIP BETWEEN TRAINING TIME AND ENCODING PROPERTIES

We next investigate the influence of training time on the encoding properties. The experiments are conducted based on one-hidden-layer MLPs with three different widths on MNIST and five-hidden-layer MLPs with four different widths on CIFAR-10. Totally, $810$ models are tested. The experiments are repeated for $5$ trials on MNIST and $10$ trials on CIFAR-10, respectively.

We collect the redundancy ratio and the categorization accuracy of every epoch in all the scenarios, as presented in Figures 3. Most results for MNIST are given in Appendix B.2 due to space limitation. The plots clearly suggest a positive correlation between the encoding properties and the training

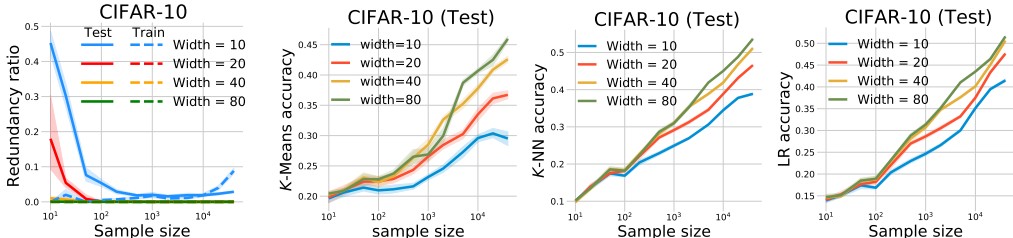

(a) R ratio vs. sample size    (b) Test accuracy of $K$-Means, $K$-NN, and logistic regression vs. sample size

Figure 4: (a) Redundancy ratios (R ratios) on training set (dotted lines) and test set (solid lines) of CIFAR-10 as functions of sample size. (b) Test accuracy of $K$-Means (left), $K$-NN (middle), and logistic regression (right) as functions of sample size. The models are MLPs of widths 10 (blue), 20 (red), 40 (orange), and 80 (green). All models are trained on CIFAR-10 for classification for 10 times with different random seeds. The darker lines show the average over seeds and the shaded area shows the standard deviations.

time: when the training time goes longer, (1) the redundancy ratio monotonically decreases; and (2) the categorization accuracy monotonically increases.

We also observe that the redundancy ratio of an untrained MLP on MNIST is almost 0; see Figure 3(a). Our explaination is as follows. When a neural network is randomly initialized, the input space is randomly partitioned into multiple activation regions. If these activation regions are sufficiently small, almost every training datum has its own activation region. However, the mapping from input data to the output prediction is meanless at random initialization, because the neural network may output two completely different predictions to two datums from neighboring activation regions. Therefore, the categorization accuracy is poor, which is consistent with your understanding. This phenomenon also suggests that only determinism is not sufficient to measure the encoding properties. It coincides with the reservoir effects (Jaeger, 2001; Maass et al., 2002).

It is worth noting that our finding is different from the result in Hanin & Rolnick (2019b) that the linear region counting increases as training progressing. We reported that the encoding properties do not apply beyond the training data distribution, no matter how the linear region counting change; see Figure 2(f). This suggests that the increase of linear region counting would just happen in a small part of the input space which is usually extremely large. Therefore, an increasing linear region counting cannot guarantee a decreasing redundancy ratio.

### 5.3 RELATIONSHIP BETWEEN SAMPLE SIZE AND ENCODING PROPERTIES

We then investigate how sample size impacts the encoding properties. We trained 210 one-hidden-layer MLPs with three different widths and 480 five-hidden-layer MLPs with four different widths on training sample sets of different sizes randomly drawn from of MNIST and CIFAR-10, respectively, while all irrelevant variables are strictly controlled. We adopt the number of iterations rather than epochs to measure the training time because the number of iterations in one epoch grows proportionally with the sample size. The experiments are repeated for 5 trials on MNIST and 10 trials on CIFAR-10, respectively

We calculated the redundancy ratio and categorization accuracy in all cases; see Figures 4(a) and 4(b), respectively. We only present here the results collected on CIFAR-10 due to space limitation. The results for MNIST are given in Appendix B.3 The plots suggest that (1) the redundancy ratio calculated on either the training sample set or the test sample starts at a considerably high position at initialization, and then decreases monotonically to near zero as the training sample size increases; and (2) the test accuracies of all the three algorithms have clear positive correlations with the sample size: the $K$-Means accuracy increases from 20% to 40%, the $K$-NN accuracy increases from 10% to 45%, and the logistic regression accuracy increases from 15% to 45%, respectively.

Surprisingly, we observe that the encoding properties on the test set are also stronger when the training sample size goes larger. Our hypothesis is as follows. Intuitively, a larger training sample

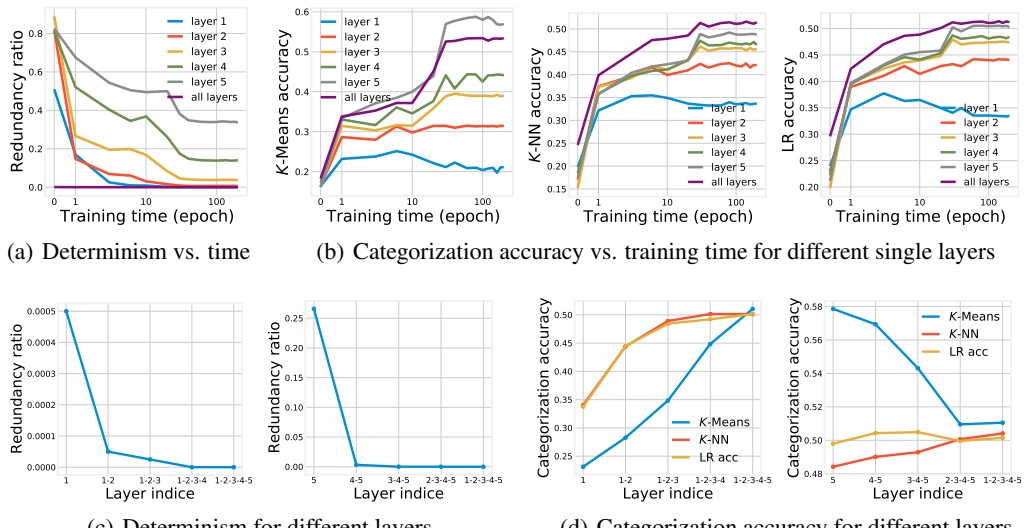

Figure 5: (a) Plots of redundancy ratio of neural codes formed by different single layers of MLPs trained on CIFAR-10 as a function of training time. (b) Test accuracy of $K$-Means (left), $K$-NN (middle), and logistic regression (right) as functions of training time. (c) Redundancy ratios of neural codes formed by multiple layers of MLPs as functions of sample size. (d) Test accuracy of $K$-Means (left), $K$-NN (middle), and logistic regression (right) as functions of sample size. The models are MLPs of width 40 on CIFAR-10.

size supports the neural network to attain a higher expressive power, i.e., the linear partition in the input space is finer. Meanwhile, a sample of larger size requires a finer linear partition to yield the same redundancy ratio. Our experiments show that the first effect is stronger than the second one. Thus, a larger sample size can help reduce the redundancy ratio.

## 5.4 LAYER-WISE ABLATION STUDY

We next study how different layers impacts the encoding properties. We conducted a layer-wise ablation study based on five-hidden-layer MLPs on the CIFAR-10 dataset, where every layer is of width 40.

We calculate the redundancy ratios and the categorization accuracy in all epochs; see Figure 5. Our results show that (1) the redundancy ratio of neural code formed by the first layer is almost always 0, while the categorization accuracy is relative poor; (2) the redundancy ratio gradually increases while the categorization accuracy gradually goes better when we test the encoding properties of neural codes formed by higher single layers; (3) the impact of the training time on the encoding properties formed by a single layer is similar to that on the neural code formed by all layers; (4) the redundancy ratio monotonically decreases when the neural code is formed by more layers; (5) the categorization accuracy gradually increases when the neural code is formed by from the first layer gradually to the while network; (6) the previous property does not hold when the neural code is formed by from the last layer gradually to the while network; and (7) the categorization accuracy of the neural code formed by the last layer is comparable with that for the whole network, which coincides with the previous two properties. The property (2) (especially the part on the redundancy ratio) reconciles the hashing property and the good generalizability of deep learning: the data is gradually concentrated to a smaller number of cells from the first layer towards the last layer, which helps neural networks generalize.

## 5.5 IMPACT OF REGULARIZATION, RANDOM DATA, AND RANDOM LABELS

We also studied the impact of regularization on the encoding properties. We trained 345 MLPs on the MNIST dataset with or without batch normalization, gradient clipping, and weight decay. The

(a) Impact of different regularizers on determinism    (b) Impact of different regularizers on LR accuracy

Figure 6: (a) Scatter of redundancy ratios of MLPs trained on MNIST with (w/, $y$-axis) or without (w/o, $x$-asix) batch normalization (BN, left), gradient clipping (middle), or weight decay (right). (b) Scatter of logistic regression (LR) accuracy of MLPs trained on MNIST with (w/, $y$-axis) or without (w/o, $x$-asix) batch normalization (BN, left), gradient clipping (middle), or weight decay (right). Every point is drawn from a model. Totally, $345$ models are involved.

results suggest that regularization has an impact on the encoding properties but relatively smaller than model size, training time, or sample size; see Figure 6. We omit the accuracies of $K$-Means and $K$-NN to the appendices due to space limitation.

We also generated random data via a uniform distribution and trained MLPs and CNNs on it. Unfortunately, the training does not converge. We then added label noise with different noise rates ($0.1$, $0.2$, $0.3$) to MNIST. The encoding properties still stand though become relatively worse. Our results suggest that the structure of the input-data can drive the organization of the hashed space. Please refer to Table 5 in Appendix B.5 and Figure 12 in Appendix B.6 for more details.

### 5.6    ACTIVATION HASH PHASE CHART

We eventually can define an *activation hash phase chart* that characterizes the space expanded by redundancy ratio, categorization accuracy, model size, training time, and sample size. Summarizing the relationships discovered above, the activation hash phase chart is divided into three canonical regions: *under-expressive regime*, *critically-expressive regime*, and *sufficiently-expressive regime*; please see more details in Section 1. This chart can help us for hyper-parameter tuning, novel algorithm designing, and algorithms diagnosis. We would also like to note that the thresholds between the three regimes are currently unknown. Exploring them is a promising future direction.

## 6    CONCLUSION

This paper studies the linear partition in the linear spaces of neural networks with ReLU-like activations. In this partition, every region corresponds to an activation pattern of the ReLU-like activations, which is parameterized by *neural code* in this paper. We discover that the neural code behaves as the hash code of the corresponding example. Specifically, the neural code possesses the following encoding properties: (1) *determinism*: almost every linear region contain at most one example. Correspondingly, almost every example can be represented by one unique neural code. This property can be quantitatively evaluated by *redundancy ratio*, which is defined to be the proportion of the examples sharing a neural code with others; and (2) *categorization*: simple classification and clustering algorithms, such as $K$-NN, logistic regression, and $K$-Means can achieve fairly good training accuracy and test accuracy on the neural code space. These properties also exhibit variabilities in different scenarios. We then find that *model size*, *training time*, *training sample size*, *regularization*, and *label noise* contribute in shaping the encoding properties, while the impacts of the first three are dominant. Accordingly, we define an *activation hash phase chart* to represent the space spanned by model size, training time, sample size, and the encoding properties, which is divided into three canonical regions: under-expressive regime, critically-expressive regime, and sufficiently-expressive regime.

### ACKNOWLEDGEMENTS

The authors sincerely appreciate the anonymous reviewers for their constructive comments, which contributed significantly in improving the quality of this paper.

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

## A   ADDITIONAL EXPERIMENT IMPLEMENTATION DETAILS

**Dataset.** Our experiments are based on the MNIST dataset (LeCun et al., 1998) and CIFAR-10 dataset (Krizhevsky & Hinton, 2009): (1)MNIST has $60,000$ training examples and $10,000$ test examples are from 10 classes. One can download this dataset at http://yann.lecun.com/exdb/mnist/; (2) CIFAR-10 is consisted of $50,000$ training images and $10,000$ test images which belong to 10 classes. One can download CIFAR-10 at https://www.cs.toronto.edu/ kriz/cifar.html. The splits of training and test sets follow the official versions. All the images are normalized so that every pixel value is in $[0, 1]$.

**Training settings.** (1) For MNIST: MLPs are trained by Adam for $2,000$ epochs with batch size of 128 and constant learning rate. VGG, ResNets, ResNeXt, and DenseNet are trained by Adam for 500 epochs with batch size of 128. Learning rate is initialed as 0.01 and decays to the $1/10$ of the previous value per 100 epochs. For all models, the hyperparameter $\beta_1$ is set as 0.9 and the hyperparameter $\beta_2$ is set to 0.999. (2) For CIFAR-10: MLPs with 5 hidden layers are trained by Adam for 200 epochs with batch size of 64. Learning rate is initialed as 0.01 and decays to the $1/10$ of the previous value per 20 epochs. VGG and ResNet are trained by SGD for 200 epoch with batch size of 64. Learning rate is initialed as 0.01 and decays to the $1/10$ of the previous value per 50 epochs. MLPs on MNIST are trained for five times with different random seeds. MLPs on CIFAR-10 are trained for ten times with different random seeds.

**Average stochastic diameter.** We first trained MLPs with width $\{5, 10, 15, 20, 25, 30, 35, 40, 45,$ $50, 55, 60, 65, 70, 75, 80, 90, 100\}$ on MNIST. Then, we randomly select 600 examples from the test set and calculate the mean of their corresponding stochastic diameters.

**Network architectures.** The network architectures involved in Section 4 are shown in the following Tables 3 and 4.

Table 3: The detailed architectures of neural networks on MNIST.

| VGG-18 | ResNet-18 | ResNet-34 | ResNeXt-26 | DenseNet-28 |
|---|---|---|---|---|
| $3 \times 3$, 32, stride 2
maxpool, $3 \times 3$ | $3 \times 3$, 32, stride 2
maxpool, $3 \times 3$ | $3 \times 3$, 32, stride 2
maxpool, $3 \times 3$ | $3 \times 3$, 32, stride 2
maxpool, $3 \times 3$ | $3 \times 3$, 6, stride 2
maxpool, $3 \times 3$ |
| $(3 \times 3, 32) \times 4$ | $\begin{bmatrix} 3 \times 3, 32 \\ 3 \times 3, 32 \end{bmatrix} \times 2$ | $\begin{bmatrix} 3 \times 3, 32 \\ 3 \times 3, 32 \end{bmatrix} \times 3$ | $\begin{bmatrix} 1 \times 1, 32 \\ 3 \times 3, 32, C = 8 \\ 1 \times 1, 64 \end{bmatrix} \times 2$ | $\begin{bmatrix} 1 \times 1, 12 \\ 3 \times 3, 3 \end{bmatrix} \times 4$ |
| $(3 \times 3, 64) \times 4$ | $\begin{bmatrix} 3 \times 3, 64 \\ 3 \times 3, 64 \end{bmatrix} \times 2$ | $\begin{bmatrix} 3 \times 3, 64 \\ 3 \times 3, 64 \end{bmatrix} \times 4$ | | $conv, 1 \times 1$
$avgpool, 2 \times 2$ |
| $(3 \times 3, 128) \times 4$ | $\begin{bmatrix} 3 \times 3, 128 \\ 3 \times 3, 128 \end{bmatrix} \times 2$ | $\begin{bmatrix} 3 \times 3, 128 \\ 3 \times 3, 128 \end{bmatrix} \times 6$ | $\begin{bmatrix} 1 \times 1, 64 \\ 3 \times 3, 64, C = 8 \\ 1 \times 1, 128 \end{bmatrix} \times 3$ | $\begin{bmatrix} 1 \times 1, 12 \\ 3 \times 3, 3 \end{bmatrix} \times 4$ |
| $(3 \times 3, 256) \times 4$ | $\begin{bmatrix} 3 \times 3, 256 \\ 3 \times 3, 256 \end{bmatrix} \times 2$ | $\begin{bmatrix} 3 \times 3, 256 \\ 3 \times 3, 256 \end{bmatrix} \times 3$ | | $conv, 1 \times 1$
$avgpool, 2 \times 2$ |
| | | | $\begin{bmatrix} 1 \times 1, 128 \\ 3 \times 3, 128, C = 8 \\ 1 \times 1, 256 \end{bmatrix} \times 3$ | $\begin{bmatrix} 1 \times 1, 12 \\ 3 \times 3, 3 \end{bmatrix} \times 4$ |
| avgpool | avgpool | avgpool | avgpool | avgpool |
| fc-10, softmax | fc-10, softmax | fc-10, softmax | fc-10, softmax | fc-10, softmax |

**Experimental designing for $K$-Means.** The pipeline for the experiments on $K$-Means is as follows: (1) we set $K$ as the number of classes; (2) run $K$-Means on the neural codes and obtain $K$ clusters; (3) every cluster can be assigned a label from $\{1, 2, \ldots, 10\}$. Thus, there are 90 (cluster, label) pairs; (4) for every (cluster, label) pair, we assign the label to all datums from the cluster and calculate the accuracy; and (5) we select the highest accuracy as the accuracy of the $K$-Means algorithm.

Table 4: The detailed architectures of neural networks on CIFAR-10.

| VGG-19 | ResNet-18 | ResNet-20 | ResNet-32 |
|---|---|---|---|
| $(3 \times 3, 32) \times 2$ 
 maxpool, $2 \times 2$ | $3 \times 3, 64$ | $3 \times 3, 16$ | $3 \times 3, 16$ |
| $(3 \times 3, 128) \times 2$ 
 maxpool, $2 \times 2$ | $\begin{bmatrix} 3 \times 3, 64 \\ 3 \times 3, 64 \end{bmatrix} \times 2$ | $\begin{bmatrix} 3 \times 3, 16 \\ 3 \times 3, 16 \end{bmatrix} \times 3$ | $\begin{bmatrix} 3 \times 3, 16 \\ 3 \times 3, 16 \end{bmatrix} \times 5$ |
| $(3 \times 3, 256) \times 4$ 
 maxpool, $2 \times 2$ | $\begin{bmatrix} 3 \times 3, 128 \\ 3 \times 3, 128 \end{bmatrix} \times 2$ | $\begin{bmatrix} 3 \times 3, 32 \\ 3 \times 3, 32 \end{bmatrix} \times 3$ | $\begin{bmatrix} 3 \times 3, 32 \\ 3 \times 3, 32 \end{bmatrix} \times 5$ |
| $(3 \times 3, 512) \times 4$ 
 maxpool, $2 \times 2$ | $\begin{bmatrix} 3 \times 3, 256 \\ 3 \times 3, 256 \end{bmatrix} \times 2$ | $\begin{bmatrix} 3 \times 3, 64 \\ 3 \times 3, 64 \end{bmatrix} \times 3$ | $\begin{bmatrix} 3 \times 3, 64 \\ 3 \times 3, 64 \end{bmatrix} \times 5$ |
| $(3 \times 3, 512) \times 4$ 
 maxpool, $2 \times 2$ | $\begin{bmatrix} 3 \times 3, 512 \\ 3 \times 3, 512 \end{bmatrix} \times 2$ | | |
| $fc - 4096$ 
 $fc - 4096$ | avgpool | avgpool | avgpool |
| fc-10, softmax | fc-10, softmax | fc-10, softmax | fc-10, softmax |

**Experiments concerning the relationship between model size and encoding properties.** We trained MLPs of widths $\{3, 7, 10, 15, 20, 23, 27, 30, 33, 37, 40, 43, 47, 50, 53, 57, 60, 65, 70, 75, 80, 90, 100\}$ on MNIST and $\{10, 20, 30, 40, 50, 60, 70, 80, 90, 100, 110, 120, 130, 140, 150, 160, 170, 180, 190, 200\}$. on CIFAR-10

**Experiments concerning the relationship between training process and model size.** (1) For MNIST, We trained MLPs with widths of $\{40, 50, 60\}$. Redundancy ratio and the test accuracy of $K$-Means, $K$-NN, and logistic regression are calculated when the training epoch is in the list of $\{1, 3, 6, 10, 30, 60, 100, 300, 600, 1000, 1200, 1500, 1800, 2000\}$. (2) For CIFAR-10, We trained MLPs with widths of $\{10, 20, 40, 80\}$. Redundancy ratio and the test accuracy of $K$-Means, $K$-NN, and logistic regression are calculated when the training epoch is in the list of $\{1, 3, 6, 10, 20, 30, 40, 60, 80, 100, 120, 140, 160, 180, 200\}$

**Experiments concerning the relationship between sample size and model size.** (1) For MNIST: We trained MLPs with widths of $\{40, 50, 60\}$ on training sample sets of size $\{10, 30, 60, 100, 300, 600, 1000, 2000, 3000, 6000, 10000, 20000, 30000, 60000\}$ randomly drawn from the training set. (2) For CIFAR-10: We trained MLPs with widths of $\{10, 20, 40, 80\}$ on training sample sets of size $\{10, 20, 50, 100, 200, 500, 1000, 2000, 5000, 10000, 20000, 40000\}$ randomly drawn from the training set.

**Experiments concerning the relationship between regularization and model size.** Three regularizers are involved in our experiments:

- Batch normalization: we add a batch normalization layer before every ReLU layer.
- Weight decay: we utilize $L_2$ weight regularizer with hyperparameter $\lambda = 0.01$.
- Gradient clipping: we set clip norm as 1.

**Layer-wise ablation study.** We trained MLPs with width of 40 on CIFAR-10. The training strategy is the same as the one previously used on MLPs with CIFAR-10.

**Experiments concerning random data** All of the pixels of random data are generalized from the uniform distribution $U(0, 1)$, individually. The shape of random example is $28 \times 28$, i.e., the same as MNIST images.

**Experiments concerning noisy labels.** Specified number of training examples of MNIST are assigned random labels according to the label noise ratios of 0.1, 0.2, and 0.3, respectively. Then, we

trained one-hidden-layer MLPs of widths $\{3, 7, 10, 15, 20, 23, 27, 30, 33, 37, 40, 43, 47, 50, 53, 57, 60, 65, 70, 75, 80, 90, 100\}$ on the noisy training set.

## B  ADDITIONAL EXPERIMENTAL RESULTS

This appendix collect experimental results omitted from the main text due to the space limitation.

### B.1  ADDITIONAL RESULTS FOR THE DIAMETERS

The following figure is for the study on the diameters. Please refer to Section 5.1.

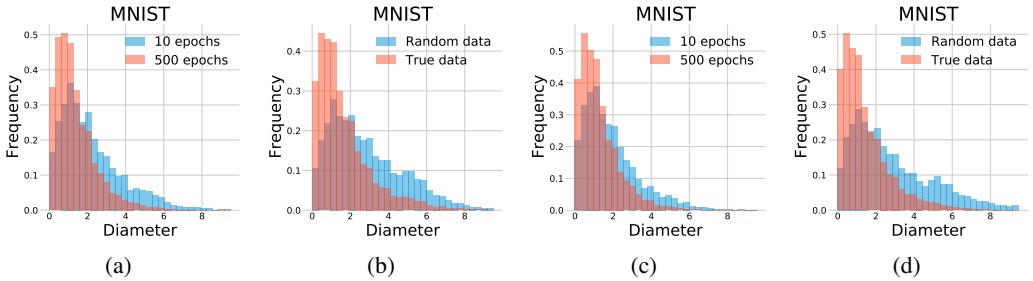

(a)  (b)  (c)  (d)

Figure 7: (a) Histograms of diameters of stochastic diameters calculated on MNIST for an MLP of width 60 trained on MNIST for 10 epochs (red) and 500 epochs (blue), respectively. (b) Histograms of stochastic diameters calculated on MNIST (blue) and randomly generated data with the same dimension (red), respectively. The model is an MLP of width 60 trained on MNIST. (c) Histograms of diameters of stochastic diameters calculated on MNIST for an MLP of width 70 trained on MNIST for 10 epochs (red) and 500 epochs (blue), respectively. (d) Histograms of stochastic diameters calculated on MNIST (blue) and randomly generated data with the same dimension (red), respectively, The model is an MLP of width 70 trained on MNIST.

## B.2 Additional results for training time

The following figure shows the encoding properties about training time on MNIST. Please refer to Section 5.2

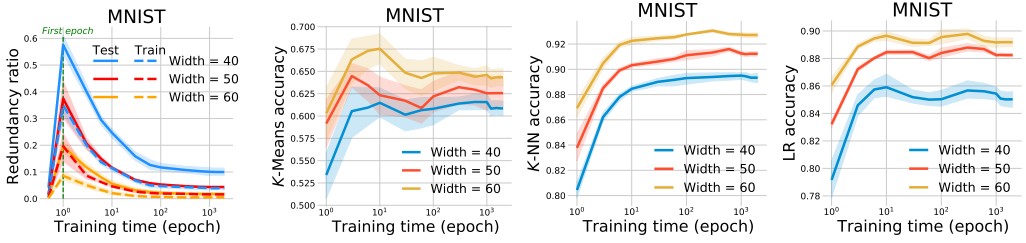

(a) Determinism vs. time    (b) Test accuracy of $K$-Means, $K$-NN, and logistic regression vs. time

Figure 8: (a) Redundancy ratio of MNIST as a function of training time. (b) Test accuracy of $K$-Means (left), $K$-NN (middle), and logistic regression (right) as functions of training time. The models are MLPs of depths $40$ (blue), $50$ (red), and $60$ (orange) on MNIST. The dotted lines show networks trained on unaltered data, evaluated with random data. All models are trained on MNIST for classification for $5$ times with different random seeds. The darker lines show the average over seeds and the shaded area shows the standard deviations.

## B.3 Additional results for sample size

The following figure shows the encoding properties about sample size on MNIST. Please refer to Section 5.3

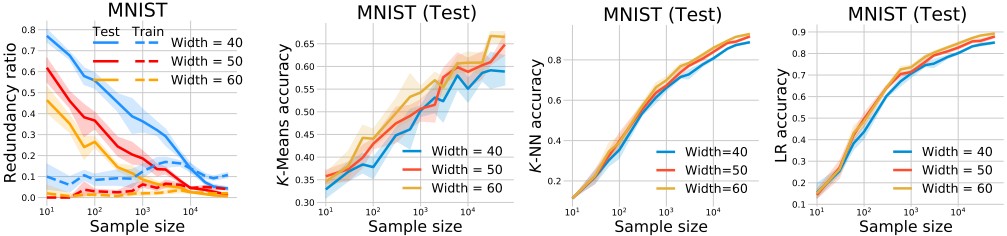

(a) Determinism vs. sample size on both training and test sets    (b) Test accuracy of $K$-Means, $K$-NN, and logistic regression vs. sample size

Figure 9: (a) Redundancy ratios on training set (dotted lines) and test set (solid lines) of MNIST as functions of sample size. (b) Test accuracy of $K$-Means (left), $K$-NN (middle), and logistic regression (right) as functions of sample size. The models are MLPs of widths $40$ (blue), $50$ (red), and $60$ (orange) on MNIST. All models are trained on MNIST for classification for $5$ times with different random seeds. The darker lines show the average over seeds and the shaded area shows the standard deviations.

## B.4 ADDITIONAL RESULTS FOR REGULARIZATION

The following figure shows the impacts of regularizers, gradient clipping and weight decay, on the redundancy ratio and the test accuracy of $K$-Means, $K$-NN, and the logistic regression.

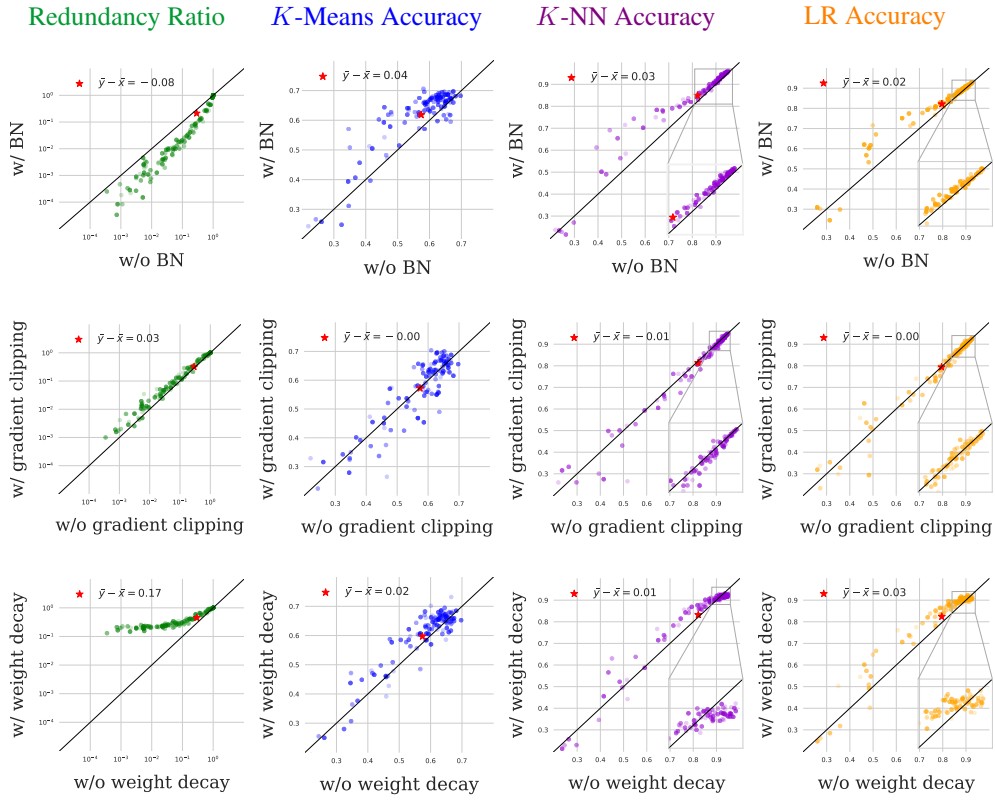

Figure 10: **First row**: Scatter of redundancy ratios and test accuracy of $K$-Means (blue), $K$-NN (violet), and logistic regression (LR, orange) for MLPs of depth from 3 to 100 with batch normalization ($y$-axis) and without gradient clipping ($x$-asix). We perfer a smaller $\bar{y} - \bar{x}$ in redundancy ration, and larger ones in the test accuracies of $K$-Means, $K$-NN, and logistic regression. Totally, 115 models are involved in one scatter. **Second row**: Scatter of redundancy ratios and test accuracy of $K$-Means (blue), $K$-NN (violet), and logistic regression (LR, orange) for MLPs of depth from 3 to 100 with gradient clipping ($y$-axis) and without gradient clipping ($x$-asix). We perfer a smaller $\bar{y} - \bar{x}$ in redundancy ration, and larger ones in the test accuracies of $K$-means, $K$-NN, and logistic regression. Totally, 115 models are involved in one scatter. **Third row**: Scatter of redundancy ratios and test accuracy of $K$-Means (blue), $K$-NN (violet), and logistic regression (LR, orange) for MLPs of depth from 3 to 100 with weight decay ($y$-axis) and without weight decay ($x$-asix). We perfer a smaller $\bar{y} - \bar{x}$ in redundancy ration, and larger ones in the test accuracies of $K$-Means, $K$-NN, and logistic regression. Totally, 115 models are involved in one scatter.

## B.5 ADDITIONAL RESULTS FOR RANDOM DATA

Figure B.5 is an example of random data and every pixel of it is generated from the uniform distribution $U(0, 1)$. Table 5 reveals a one-hidden layer MLP with width $100$ fails to fit random data.

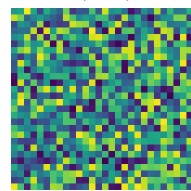

Figure 11: An example of random data

Table 5: Training accuracy and loss on training random data with one-hidden-layer MLPs

| Epoch | 0 | 100 | 300 | 500 |
|---|---|---|---|---|
| Training acc (%) | 10.92 | 11.24 | 11.24 | 11.24 |
| Loss | 230.56 | 230.13 | 230.13 | 230.13 |

## B.6 ADDITIONAL RESULTS FOR RANDOM LABEL

The following figures show the impacts of random label on redundancy ratio and the test accuracy of $K$-Means, $K$-NN, and logistic regression. Please refer to Section 5.5.

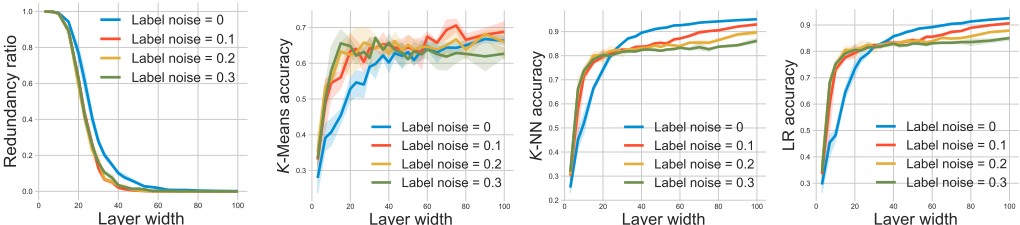

(a) Determinism in different label noises

(b) Test accuracy of $K$-Means, $K$-NN, and logistic regression in different label noises

Figure 12: (a) Redundancy ratios of MNIST as functions of layer width in different label noises. (b) Test accuracy of $K$-Means (left), $K$-NN (middle), and logistic regression (right) as functions of layer width in different label noises. The models are MLPs trained in different label noises $0$ (blue), $0.1$ (red), $0.2$ (orange) and $0.3$ (green) on MNIST. All models are trained on MNIST with noise labels for classification for $5$ times with different random seeds. The darker lines show the average over seeds and the shaded area shows the standard deviations.

