# OpenReview forum: "Neural networks behave as hash encoders: An empirical study"
_ICLR.cc/2021/Conference — Reject_

### Official Review · AnonReviewer4 · 2020-10-25
**Hash encoders of Neural networks**

**Rating:** 6
**Confidence:** 2

**Review:**

In this paper, the problem of linear partition in the linear spaces of neural networks with ReLU-like activations is studied. It demonstrates that such a partition exhibits two properties, including determinism and categorization, across a variety of deep learning models

————————————————————————————————————————————

This paper presents an interesting work, however, there are a few issues/comments with the work:

1.This paper mentions that “simple classification and clustering algorithms, such as K-NN, logistic regression, and K-means can achieve fairly good training accuracy and test accuracy on the neural code space”,  About “achieve fairly good training accuracy and test accuracy”, maybe some comparisons with existing methods will strength this statement.

2.The paper concludes that model capacity, training time, and sample size play important roles in shaping the encoding properties, while several popular regularizers have little impact on the encoding properties.  As for the conclusion of the regularization technique, it is better to try other methods, such as drop-out, to verify gradient clipping and weight decay not only on MLP but also on RNN, before reaching such a conclusion. Moreover, only for MLP on MNIST, it seems unconvincing.  It will be better to try out some existing/well-known NN models on more datasets.
3.I was confused about the sentence below Section 5.4 “we trained 345 MLPs on the MNIST dataset” and the last sentence in the caption of Figure6 “Totally, 115 models are involved in one scatter. “. If possible, could you please explain that?

4.In Figure 2 (c) and (d), “Diameter“ on the horizontal axis correspondings to the “Layer width” in Figure 2 (a) and (b)

5.What is the meaning of “Frequency” on the vertical axis?

6.In Figure 6, the third and fourth subfigures mean that for some of MLPs with depths between 3 and 100, “ without BN” would be better than  “with BN”, BN sometimes harms the performance of these models? is that so?

7.It needs some space between the second and the third paragraphs below Figure 2.

8.In deep models, depth generally is more important than width. Why only analyze the model capacity by width?

9.For the definitions of three regions “	Under-expressive regime”, “Critically-expressive regime”, and “Sufficiently-expressive regime”, may not need to be repeated on page 8, as similar definitions are explained on page 2. Also, is that possible to quantify/formalize them? If so, it would be more interesting and useful.

10.The format of references is inconsistent, for example, references “Jingdong Wang, Ting Zhang, Nicu Sebe, Heng Tao Shen, et al.. 2017” and “Li Yuan, Eng Hock Francis Tay, Ping Li, and Jiashi Feng. 2019”.

==========================================================================================

"In addition, I am not sure the description would be enough to reproduce and no code seems to provide." In the beginning, I did not find the codes related to this paper, but later, the author(s) uploaded codes. Thanks.  For the codes,  if setting random seed program-wide in codes maybe it will be more helpful for reproducing.

==========================================================================================

————————————————————————————————————————————

Overall, I think this is an interesting piece of work that may be of interest to people to explore/understand (deep) neural networks, but I think the results/conclusions require more careful/extended analysis.

---

> ### Author Response · Authors · 2020-11-14
> **To Reviewer #4**
>
> Thank you very much for your thorough review, helpful feedback, and kind support! We have carefully responded to your questions and modified the manuscript accordingly.
>
> Q1: _This paper mentions that “simple classification and clustering algorithms, such as K-NN, logistic regression, and K-means can achieve fairly good training accuracy and test accuracy on the neural code space”, About “achieve fairly good training accuracy and test accuracy”, maybe some comparisons with existing methods will strength this statement._
>
> A1: Thanks. To our best knowledge, this is the first work that presents and studies the encoding properties exhibited in the activation patterns. We will give more discussion concerning the test accuracy. The accuracies of these simple algorithms on the neural codes are comparable to the test accuracy of the corresponding network on the raw images.
>
> Q2.1: _The paper concludes that model capacity, training time, and sample size play important roles in shaping the encoding properties, while several popular regularizers have little impact on the encoding properties. As for the conclusion of the regularization technique, it is better to try other methods, such as drop-out, to verify gradient clipping and weight decay not only on MLP but also on RNN, before reaching such a conclusion._
>
> A2.1: Thanks. We respectfully note that we have also conducted experiments for gradient clipping and weight decay. The results were given in the appendix and have been moved to the main text.
>
> Q2.2: _Moreover, only for MLP on MNIST, it seems unconvincing. It will be better to try out some existing/well-known NN models on more datasets._
>
> A2.2: Thanks. We conducted more experiments of MLPs, VGGs, and ResNets on CIFAR-10. Please refer to our post to all reviewers and the “Additional experimental results.pdf” in the supplementary material for more details.
>
> Q3: _I was confused about the sentence below Section 5.4 “we trained 345 MLPs on the MNIST dataset” and the last sentence in the caption of Figure 6 “Totally, 115 models are involved in one scatter.” If possible, could you please explain that?_
>
> A3: Thanks. We conducted experiments for three regularizers, batch normalization, gradient clipping, and weight decay. Each of them involves $115$ MLPs. Totally, $345$ MLPs are involved.  Figure 6 is for batch normalization.
>
> Q4: _In Figure 2 (c) and (d), “Diameter“ on the horizontal axis correspondings to the “Layer width” in Figure 2 (a) and (b)._
>
> A4: Yes, we agree.
>
> Q5: _What is the meaning of “Frequency” on the vertical axis?_
>
> A5: Thanks. “Frequency” is the proportion of the diameters that fall in a specific range. For example, the frequency for the range $(0, 1)$ is the proportion of all the tested linear regions whose diameter falls in the range $(0, 1)$.
>
> Q6: _In Figure 6, the third and fourth subfigures mean that for some of MLPs with depths between 3 and 100, “ without BN” would be better than “with BN”, BN sometimes harms the performance of these models? is that so?_
>
> A6: Yes, we agree.
>
> Q7: _It needs some space between the second and the third paragraphs below Figure 2._
>
> A7: Thanks and addressed.
>
> Q8: _In deep models, depth generally is more important than width. Why only analyze the model capacity by width?_
>
> A8: Thanks. We trained four ResNets with different depths on CIFAR-10 but do not observe a significant discrepancy in the encoding properties. We hypothesize it is because the optimal training protocol (especially training time) for networks of different depths significantly differs. Thus, it is hard to conduct experiments on depth while controlling other factors.
>
> Q9: _For the definitions of three regions “ Under-expressive regime”, “Critically-expressive regime”, and “Sufficiently-expressive regime”, may not need to be repeated on page 8, as similar definitions are explained on page 2. Also, is that possible to quantify/formalize them? If so, it would be more interesting and useful._
>
> A9: Thank you for your suggestion. We agree that it is very interesting to draw a diagram and find the quantitative thresholds between the three regimes. They can help us for hyper-parameter tuning, novel algorithm designing, and algorithms diagnosis. We are keen to explore in this direction. However, such exploration requires large-scale experiments, but our current computing resources do not support us to complete these experiments. We are applying for sufficient computing resources to complete this exploration.
>
> Q10: _The format of references is inconsistent, for example, references “Jingdong Wang, Ting Zhang, Nicu Sebe, Heng Tao Shen, et al.. 2017” and “Li Yuan, Eng Hock Francis Tay, Ping Li, and Jiashi Feng. 2019”._
>
> A10: Thanks and addressed.
>
> Q11: _In addition, I am not sure the description would be enough to reproduce and no code seems to provide._
>
> A11: Thanks. We have submitted our source code package as supplementary material. Our code, trained models, and collected data will all be released publicly later.

---

### Official Review · AnonReviewer3 · 2020-10-25
**Interesting empirical observation that intuitively makes sense - but currently lacking some important controls.**

**Rating:** 7
**Confidence:** 4

**Review:**

**Update after authors' response**
I am very happy to see the additional results on CIFAR-10, and the layer-wise ablations and other control experiments. To me, these results have shed a lot of light onto how the observed hashing effect can be explained. These explanations mostly confirm intuitions. Nonetheless I think it's worth reporting the empirical verification of these intuitions that tie in with earlier results reported in other papers. To me, the most interesting aspect of the work is how the hashing properties change over training (and across layers). The paper could still be improved by experiments on e.g. CIFAR-100, ImageNet and non-vision tasks, as well as more mathematically sophisticated definitions of some of the measures (e.g. average stochastic activation diameter). I personally think the results are now sufficient (and sufficiently backed up) for a publication and most of the criticism raised by the other reviewers has been addressed sufficiently (for me). I would now rate the paper as a 6.5 - but to facilitate the reviewer's discussion I will take a clear stance and have thus raised my score to 7.
---

**Summary**
The paper investigates activation patterns in ReLU networks, which are known to be piecewise linear. The main finding is that trained MNIST classifiers produce unique activation patterns for most points from the data-distribution (but not for random data). To be precise, an activation pattern is a binary matrix where each entry corresponds to one ReLU unit, and is 1 if the unit has non-zero activation and 0 otherwise. This means that trained MNIST classifiers can be viewed as “hashing” each data-point into a unique activation pattern. Importantly, simple clustering/classification algorithms (K-means, K-nearest-neighbors, and logistic regression) on these hashed patterns lead to good classification accuracy, implying that the hashed patterns follow some underlying geometric regularity (which is not trivially expected from an arbitrary hashing function).

---------------------------------------------------------------------
**Main contributions, Novelty, Impact**
1) Systematic study of the “hashing” behavior of MNIST classifiers. To the best of my knowledge this particular type of analysis is novel. The findings are interesting, though perhaps not totally surprising. The impact of the results is currently limited by 2 important factors: (i) results are shown on MNIST only, and (ii) it is currently unclear whether the hashing effect is mostly explained by early layer activations, or not. To address (ii) layer-wise ablations would be crucial (see improvements for more details).

2) Various ablations and control experiments to identify the impact of certain hyper-parameters and architectural variations. The ablations and control experiments are interesting to further characterize the observed hashing effect. Two important control experiments are currently missing: (i) results for untrained networks, and (ii) results when fitting and random labels (to determine whether the observed effects are mainly input-data-driven, or whether the hashed landscape is mainly shaped by label information).

3) Definition of an “Activation Hash Phase Chart” with three qualitatively distinct reasons. The main idea behind this is interesting, but the paper never shows such a diagram, and (importantly!) how the chart is meant to be used (hyper-parameter tuning, or rather as a diagnostic tool, …?). It is thus currently unclear whether such a chart could have wider impact in the community.

---------------------------------------------------------------------
**Score and reasons for score**
The main observation in the paper is interesting and supported by a number of experiments. Unfortunately the generality of the findings is currently fairly limited since evaluation happened for MNIST classification only, and some control experiments are missing. I think the ingredients for a strong and interesting paper are there, but it does not quite come together yet. I therefore currently suggest a major revision of the work, which means resubmission at another conference. I personally want to strongly encourage the authors to spend a bit more time and work on the paper to turn it into the strongest version possible - in which case I think the paper could be quite impactful.  I am of course happy to reconsider my final verdict based on the other reviews and the authors' response.

Perhaps the most important unaddressed point is how the observed hashing property can be reconciled with the good generalization performance: if each test-datapoint is mapped to a unique pattern, how does the classifier generalize well? I suspect that the hashing property is mostly a property of early layers that gets washed out towards the output of the classifier (but it might also be the case that the hashing property holds essentially until the final layer). I think the hashing property itself is not very surprising, but what’s surprising is that the “goodness-of-hash” changes quite a bit over training - not only for training points but also test data, but not random data. Shedding more light on this would make the paper much stronger in my opinion.

---------------------------------------------------------------------
**Strengths**
 * Observed hashing effect and hypothesis stated fits quite well with the wider literature (which is also well cited in the paper).
 * The experiments that are conducted are thorough (multiple repetitions per run, covering a broad range of hyperparameters).
 * To me Fig 3c is the most convincing result that the hashing effect is (at least partly) an caused by training and not simply a generic feature neural networks with a certain width.
---------------------------------------------------------------------
**Weaknesses**
 * Results are on MNIST only. Historically it’s often been the case that strong results on MNIST would not carry over to more complex data. Additionally, at least some core parts of the analysis does not require training networks (but could even be performed e.g. with pre-trained classifiers on ImageNet) - there is thus no severe computational bottleneck, which is often the case when going beyond MNIST.
 * The “Average stochastic activation diameter” is a quite crude measure and results must thus be taken with a (large) grain of salt. It would be good to perform some control experiments and sanity checks to make sure that the measure behaves as expected, particularly in high-dimensional spaces.
 * The current paper reports the hashing effect and starts relating it to what’s known in the literature, and has some experiments that try to understand the underlying *causes* for the hashing effect. However, while some factors are found to have an influence on the strength of the effect, some control experiments are still missing (training on random labels, results on untrained networks, and an analysis of how the results change when starting to leave out more and more of the early layers).

---------------------------------------------------------------------
**Correctness**
Overall the methodology, results, and conclusions seem mostly fine (I’m currently not very convinced by the “stochastic activation diameter” and would not read too much into the corresponding results). Additionally some claims are not entirely supported (in fullest generality), based on the results shown, see comments for more on this.

---------------------------------------------------------------------
**Clarity**
The main idea is well presented and related literature is nicely cited. However, some of the writing is quite redundant (some parts of the intro appear as literal copies later in the text). Most importantly the writing in some parts of the manuscript seems quite rushed with quite a few typos and some sentences/passages that could be rephrased for more fluent reading.

---------------------------------------------------------------------
**Improvements (that would make me raise my score) / major issues (that need to be addressed)**
1) Experiments on more complex datasets.

2) One question that is currently unresolved is: is the hashing effect mostly attributed to early layer activations? Ultimately, a high-accuracy classifier will “lump together” all datapoints of a certain class when looking at the network output only. The question is whether this really happens at the very last layer or already earlier in the network. Similarly, when considering the input to the network (the raw data) the hashing effect holds since each data-point is unique. It is conceivable that the first layer activations only marginally transform the data in which case it would be somewhat trivially expected to see the hashing effect (when considering all activations simultaneously). However that might not explain e.g. the K-NN results.
I think it would be very insightful to compute the redundancy ratio layer-wise and/or when leaving out more and more of the early layer activations (i.e. more and more rows of the activation pattern matrix). Additionally it would be great to see how this evolves over time, i.e. is the hashing effect initially mostly localized in early layers and does it gradually shape deeper activations over training? This would also shed some light on the very important issue of how a network that maps each (test-) data-point to a unique pattern generalize well?

3) Another unresolved question is whether it’s mostly the structure of the input-data or the labels driving the organization of the hashed space? The random data experiments answers this partially. Additionally it would be interesting to see what happens when (i) training with random data, (ii) training with random labels - is the hashing effect still there, does the K-NN classification still work?

4) Clarify: Does Fig 3c and 4a show results for untrained networks? I.e. is the redundancy ratio near 0 for training, test and random data in an untrained network? I would not be entirely surprised by that (a “reservoir effect”) but if that’s the case that should be commented/discussed in the paper, and improvement 3) mentioned above would become even more important. If the figures do not show results for untrained networks then please run the corresponding experiments and add them to the figures and Table 1.

5) Clarify: Random data (Fig 3c). Was the network trained on random data, or do the dotted lines show networks trained on unaltered data, evaluated with random data?

6) Clarify: Random data (Fig 3). Was the non-random data normalized or not (i.e. is the additional “unit-ball” noise small or large compared to the data). Ideally show some examples of the random data in the appendix.

7) P3: “It is worth noting that the volume of boundaries between linear regions is zero” - is this still true for non-ReLU nonlinearities (e.g. sigmoids)? If not what are the consequences (can you still easily make the claims on P1: “This linear region partition can be extended to the neural networks containing smooth activations”)? Otherwise please rephrase the claims to refer to ReLU networks only.

8) I disagree that model capacity is well measured by layer width. Please use the term ‘model-size’ instead of ‘model-capacity’ throughout the text. Model capacity is a more complex concept that is influenced by regularizers and other architectural properties (also note that the term capacity has e.g. a well-defined meaning in information theory, and when applied to neural networks it does not simply correspond to layer-width).

9) Sec 5.4: I disagree that regularization “has very little impact” (as mentioned in the abstract and intro). Looking at the redundancy ratio for weight decay (unfortunately only shown in the appendix) one can clearly see a significant and systematic impact of the regularizer towards higher redundancy ratios (as theoretically expected) for some networks (I guess the impact is stronger for larger networks, unfortunately Fig 8 in the appendix does not allow to precisely answer which networks are which).

---------------------------------------------------------------------
**Minor comments**
A) Formally define what “well-trained” means. The term is used quite often and it is unclear whether it simply means converged, or whether it refers to the trained classifier having to have a certain performance.

B) There is quite an extensive body of literature (mainly 90s and early 2000s) on “reservoir effects” in randomly initialized, untrained networks (e.g. echo state networks and liquid state machines, however the latter use recurrent random nets). Perhaps it’s worth checking that literature for similar results.

C) Remark 1: is really only the *training* distribution meant, i.e. without the *test* data, or is it the unaltered data generating distribution (i.e. without unit-ball noise)?

D) Is the red histogram in Fig 3a and 3b the same (i.e. does Fig 3b use the network trained with 500 epochs)?

E) P2 - Sufficiently-expressive regime: “This regime involves almost all common scenarios in the current practice of deep learning”. This is a bit of a strong claim which is not fully supported by the experiments - please tone it down a bit. It is for instance unclear whether the effect holds for non-classification tasks, and variational methods with strong entropy-based regularizers, or Dropout, ...

F) P2- The Rosenblatt 1961 citation is not entirely accurate, MLP today typically only loosely refers to the original Perceptron (stacked into multiple-layers), most notably the latter is not trained via gradient backpropagation. I think it’s fine to use the term MLP without citation, or point out that MLP refers to a multi-layer feedforward network (trained via backprop).

G) First paragraph in Sec. 4 is very redundant with the first two bullet points on P2 (parts of the text are literally copied). This is not a good writing style.

H) P4 - first bullet point: “Generally, a larger redundancy ratio corresponds a worse encoding property.”. This is a quite hand-wavy statement - “worse” with respect to what? One could argue that for instance for good generalization high redundancy could be good.

I) Fig 3: “10 epochs (red) and 500 epochs (blue),” does not match the figure legend where red and blue are swapped.

J) Fig 3: Panel b says “Rondom” data.

K) Should the x-axis in Fig 3c be 10^x where x is what’s currently shown on the axis? (Similar to how 4a is labelled?)

L) Some typos
P2: It is worths noting
P2:  By contrast, our the partition in activation hash phase chart characerizes goodnessof-hash.
P3: For the brevity
P3: activation statue

---

> ### Author Response · Authors · 2020-11-14
> **To Reviewer #3 (3/3)**
>
> A11: Thanks for your suggestions! Our new layer-wise ablation study coincides with your hypothesis: the hashing property (or the determinism property) is fairly good in the early layers and then gradually goes poor towards the final layer. This reconciles the hashing property with the good generalizability of deep learning.
>
> ---
>
> **For Minor comments:**
>
> Q(A): _Formally define what “well-trained” means. The term is used quite often and it is unclear whether it simply means converged, or whether it refers to the trained classifier having to have a certain performance._
>
> A(A): Thanks. “Well-trained” means the training procedure has converged. We have added a formal definition in our manuscript.
>
> Q(B): _There is quite an extensive body of literature (mainly 90s and early 2000s) on “reservoir effects” in randomly initialized, untrained networks (e.g. echo state networks and liquid state machines, however the latter use recurrent random nets). Perhaps it’s worth checking that literature for similar results._
>
> A(B): Thank you for your suggestion! We have carefully checked the literature with the keywords “reservoir effects”, “neural networks”, “echo state networks”, and “liquid state machines”. Reservoir effects coincide with the phenomenon that untrained neural networks have near $0$ redundancy ratios. The related papers have been clearly acknowledged in our manuscript.
>
> Q(C): _Remark 1: is really only the training distribution meant, i.e. without the test data, or is it the unaltered data generating distribution (i.e. without unit-ball noise)?_
>
> A(C): Thanks and addressed. We have replaced “training data distribution” by “unaltered data generating distribution”.
>
> Q(D): _Is the red histogram in Fig 3a and 3b the same (i.e. does Fig 3b use the network trained with 500 epochs)?_
>
> A(D): Thanks. The red histograms in Fig 3a and 3b are identical. The scales of the vertical axis are different in the two histograms. We have noted this in the manuscript.
>
> Q(E): _P2 - Sufficiently-expressive regime: “This regime involves almost all common scenarios in the current practice of deep learning”. This is a bit of a strong claim which is not fully supported by the experiments - please tone it down a bit. It is for instance unclear whether the effect holds for non-classification tasks, and variational methods with strong entropy-based regularizers, or Dropout, ..._
>
> A(E): Thanks and addressed. We have toned it accordingly as below.
>
> “This regime covers many popular scenarios in the current practice of deep learning, especially those in classification”.
>
> Q(F): _P2- The Rosenblatt 1961 citation is not entirely accurate, MLP today typically only loosely refers to the original Perceptron (stacked into multiple-layers), most notably the latter is not trained via gradient backpropagation. I think it’s fine to use the term MLP without citation, or point out that MLP refers to a multi-layer feedforward network (trained via backprop)._
>
> A(F): Thanks and addressed. We have removed the citation.
>
> Q(G): _First paragraph in Sec. 4 is very redundant with the first two bullet points on P2 (parts of the text are literally copied). This is not a good writing style._
>
> A(G): Thanks and addressed. We will carefully modify this part to avoid redundancy.
>
> Q(H): _P4 - first bullet point: “Generally, a larger redundancy ratio corresponds a worse encoding property.” This is a quite hand-wavy statement - “worse” with respect to what? One could argue that for instance for good generalization high redundancy could be good._
>
> A(H): Thanks and addressed. We have modified it as follows.
>
> “Generally, a smaller redundancy ratio is preferred when we evaluating the activation mapping as a hashing function.”
>
> Q(I): _Fig 3: “10 epochs (red) and 500 epochs (blue),” does not match the figure legend where red and blue are swapped._
>
> A(I): Thanks and addressed. We have replaced it by “10 epochs (blue) and 500 epochs (red)”.
>
> Q(J): _Fig 3: Panel b says “Rondom” data._
>
> A(J): Thanks and addressed.
>
> Q(K): _Should the x-axis in Fig 3c be 10^x where x is what’s currently shown on the axis? (Similar to how 4a is labelled?)_
>
> A(K): The x-axis in Fig 3c is correct here. Thank you for noting this! We will modify Fig 3c to make the x-axis consistent.
>
> Q(L): _Some typos P2: It is worths noting P2: By contrast, our the partition in activation hash phase chart characerizes goodnessof-hash. P3: For the brevity P3: activation statue._
>
> A(L): Thanks and addressed.

---

> ### Author Response · Authors · 2020-11-14
> **To Reviewer #3 (2/3)**
>
> Q5: _Clarify: Random data (Fig 3c). Was the network trained on random data, or do the dotted lines show networks trained on unaltered data, evaluated with random data?_
>
> A5: In Fig 3c, the dotted lines show networks trained on unaltered data, evaluated with random data.
>
> Q6: _Clarify: Random data (Fig 3). Was the non-random data normalized or not (i.e. is the additional “unit-ball” noise small or large compared to the data). Ideally show some examples of the random data in the appendix._
>
> A6: Yes, the non-random data has been normalized, i.e., every pixel value is normalized to $[0, 1]$. Moreover, we will show some examples of the random data in the appendix. Please refer to the “Additional experimental results.pdf” in the updated supplementary material.
>
> Q7: _P3: “It is worth noting that the volume of boundaries between linear regions is zero” - is this still true for non-ReLU nonlinearities (e.g. sigmoids)? If not what are the consequences (can you still easily make the claims on P1: “This linear region partition can be extended to the neural networks containing smooth activations”)? Otherwise please rephrase the claims to refer to ReLU networks only._
>
> A7: Thank you. “It is worth noting that the volume of boundaries between linear regions is zero” is true if the neural network has ReLU nonlinearities no matter whether there are non-ReLU nonlinearities. This statement no longer stands if the neural network does not have ReLU nonlinearities any more. We respectfully note that the full statement is “This linear region partition can be extended to the neural networks containing smooth activations (such as sigmoid activations and tanh activations) as well as ReLU-like activations.” We will rephrase the sentence as follows to make it clearer.
>
> “This linear region partition still holds if the neural network contains smooth activations (such as sigmoid activations and tanh activations) besides ReLU-like activations, in which the interiors are no longer linear but still smooth.”
>
> Q8: _I disagree that model capacity is well measured by layer width. Please use the term ‘model-size’ instead of ‘model-capacity’ throughout the text. Model capacity is a more complex concept that is influenced by regularizers and other architectural properties (also note that the term capacity has e.g. a well-defined meaning in information theory, and when applied to neural networks it does not simply correspond to layer-width)._
>
> A8: Thanks and addressed. We have replaced all “model-capacity” by “model-size”.
>
> Q9: _Sec 5.4: I disagree that regularization “has very little impact” (as mentioned in the abstract and intro). Looking at the redundancy ratio for weight decay (unfortunately only shown in the appendix) one can clearly see a significant and systematic impact of the regularizer towards higher redundancy ratios (as theoretically expected) for some networks (I guess the impact is stronger for larger networks, unfortunately Fig 8 in the appendix does not allow to precisely answer which networks are which)._
>
> A9: Thanks and addressed. We will present the results on regularization in the main text. We will also carefully modify the statement on the regularization as below.
>
> "Regularization also has an impact on the encoding properties, but is relatively smaller than model size, sample size, or training time in our experiments."
>
> ---
>
> **For more questions in the Main contributions, Novelty, Impact part:**
>
> Q10: _Definition of an “Activation Hash Phase Chart” with three qualitatively distinct reasons. The main idea behind this is interesting, but the paper never shows such a diagram, and (importantly!) how the chart is meant to be used (hyper-parameter tuning, or rather as a diagnostic tool, …?). It is thus currently unclear whether such a chart could have wider impact in the community._
>
> A10: Thanks for recognizing our contribution. We agree that it is very interesting to draw a diagram and further find the quantitative thresholds between the three regimes. They can help us for hyper-parameter tuning, novel algorithm designing, and algorithms diagnosis, as you have mentioned. We are keen to explore in this future direction. Such exploration requires large-scale experiments, but our current computing resources do not support us to complete these experiments. We are applying for sufficient computing resources to complete this exploration.
>
> ---
>
> **For more questions in the Score and reasons for score part:**
>
> Q11: _Perhaps the most important unaddressed point is how the observed hashing property can be reconciled with the good generalization performance: if each test-datapoint is mapped to a unique pattern, how does the classifier generalize well? I suspect that the hashing property is mostly a property of early layers that gets washed out towards the output of the classifier._

---

> ### Author Response · Authors · 2020-11-14
> **To Reviewer #3 (1/3)**
>
> Thank you very much for your thorough review and very helpful comments! We have carefully responded to your questions and improved our manuscript accordingly. We sincerely hope all your concerns have been addressed.
>
> ---
>
> **For questions in the Improvements (that would make me raise my score) / major issues (that need to be addressed) part:**
>
> Q1: _Experiments on more complex datasets._
>
> A1: Thank you for your suggestions. We have conducted experiments on CIFAR-10. The results fully support our arguments. Please refer to our post to all reviewers and the updated manuscript.
>
> Q2: _One question that is currently unresolved is: is the hashing effect mostly attributed to early layer activations? Ultimately, a high-accuracy classifier will “lump together” all datapoints of a certain class when looking at the network output only. The question is whether this really happens at the very last layer or already earlier in the network. Similarly, when considering the input to the network (the raw data) the hashing effect holds since each data-point is unique. It is conceivable that the first layer activations only marginally transform the data in which case it would be somewhat trivially expected to see the hashing effect (when considering all activations simultaneously). However that might not explain e.g. the K-NN results. I think it would be very insightful to compute the redundancy ratio layer-wise and/or when leaving out more and more of the early layer activations (i.e. more and more rows of the activation pattern matrix). Additionally it would be great to see how this evolves over time, i.e. is the hashing effect initially mostly localized in early layers and does it gradually shape deeper activations over training? This would also shed some light on the very important issue of how a network that maps each (test-) data-point to a unique pattern generalize well?_
>
> A2: Thank you. We have added a layer-wise ablation study for understanding the behaviours of the neural code of some layers. The results show that the earlier layers have fairly low redundancy ratios but relatively poor categorization accuracies, while higher layers have very poor redundancy ratios though the categorization accuracies are fairly high. One can only observe both encoding properties are satisfied in neural code formed by all layers. The two properties collectively measure the expressive power of a neural network. Thus, our results suggest that the neural code formed by any part of the neural network are not sufficient compared with that formed by the whole neural network (i.e., all layers). Please refer to our post to all reviewers and the updated manuscript.
>
> Q3: _Another unresolved question is whether it’s mostly the structure of the input-data or the labels driving the organization of the hashed space? The random data experiments answers this partially. Additionally it would be interesting to see what happens when (i) training with random data, (ii) training with random labels - is the hashing effect still there, does the K-NN classification still work?_
>
> A3: Thank you. We first generate random data via a Gaussian distribution and train MLPs and CNNs on it. Unfortunately, the training does not converge any more. We then add label noise with different noise rate to MNIST. The encoding properties still stand though become relatively worse. Our results suggest that the structure of the input-data can drive the organization of the hashed space. Please refer to our post to all reviewers and the updated manuscript.
>
> Q4: _Clarify: Does Fig 3c and 4a show results for untrained networks? I.e. is the redundancy ratio near 0 for training, test and random data in an untrained network? I would not be entirely surprised by that (a “reservoir effect”) but if that’s the case that should be commented/discussed in the paper, and improvement 3) mentioned above would become even more important._
>
> A4: Yes, Figures 3(c) and 4(a) show that the redundancy ratio of an untrained network is near $0$ on training, test, and random data. We will add a detailed discussion as below.
>
> "When a neural network is randomly initialized, the input space is randomly partitioned into multiple activation regions. If these activation regions are sufficiently small, almost every training datum has its own activation region. However, the mapping from input data to the output prediction is meanless at random initialization, because the neural network may output two completely different predictions to two datums from neighboring activation regions. Therefore, the categorization accuracy is poor, which is consistent with your understanding. This phenomenon also suggests that only determinism is not sufficient to measure the encoding properties. It coincides with the “resevior effects” as you have suggested."

---

### Official Review · AnonReviewer1 · 2020-10-27

**Rating:** 6
**Confidence:** 4

**Review:**

**Update after rebuttal:** I thank the authors for their detailed responses and the additional experiments. The responses addressed most of my concerns. I noticed that I had the wrong notion of redundancy ratio in my mind (I'm glad the authors now give a more formal definition of this concept as I think this would trip up many other readers). I'm also glad that the authors have clarified the difference between their results and those reported in Hanin and Rolnick (ICML, 2019). Given these, I'm happy to increase my score to a weak accept (a weak accept, because I'm still not quite sure about the significance of the results reported in this paper).

--------------------------------------------------------------
This paper reports some observations about properties of linear regions in deep ReLU networks. Unfortunately, I have several major issues with the paper.

(1) First of all, I am a bit confused about the motivation behind this work. The motivation is initially couched in terms of hash codes, so one gets the impression that the authors are going to propose a new hash coding scheme using neural networks. But this is clearly not the case. The proposed coding scheme is practically useless as a hash code because of the enormous dimensionality of the codes (I estimate this to be on the order of millions even for the toy MNIST case studied in the paper, but it would actually be very helpful for the reader if the authors explicitly mentioned the dimensionality of the proposed hash codes--at least the order of magnitude--). This is also why the authors are stuck with the toy MNIST dataset throughout the paper, because the proposed scheme is completely impractical for any reasonably large dataset and model.

(2) This begs the question: what exactly is the significance of the observation that linear regions satisfy some properties of hash codes, if they’re not going to be used as hash codes? It’s not meaningful to just point out that something satisfies some properties of hash codes. One can point to a million different things that satisfy some properties of hash codes. What exactly is the significance of linear regions having these properties?

(3) This brings me to the related works section. This section is written very shallowly, the authors do not do a good job of situating their contributions in the context of prior works. You cannot just say: “Other advances in linear region counting include …” (p. 3) and then cite a bunch of references. You have to tell us what each of these papers did and how what you’re doing in this paper differs from these earlier works and makes a meaningful and novel contribution to the prior literature.

(4) The concept of redundancy ratio seems to play a central role in the paper, but it is not defined formally, just a verbal (potentially ambiguous) definition is given. Please define this concept formally to avoid any ambiguities. I’m also not sure this concept is the right one for quantifying the goodness of a code: consider a case where each point in a dataset shares its linear region with exactly one other point in the dataset vs. a case where all points belong to the same linear region. It seems that the redundancy ratio will be 100% in both cases, however intuitively the code in the first case should be much better than in the second case (for example for retrieval).

(5) Relatedly, Figure 3c suggests that the redundancy ratio is close to zero for an untrained random network. Then, by the authors’ own definition, the encoding is actually very good before training. Please consider what this means (also see point 7 below).

(6) Unfortunately, I don’t think classification results for MNIST only are very meaningful. Almost anything will get above 99% accuracy on MNIST. Moreover, no effort is made by the authors to understand what drives good test accuracy in these experiments. A very straightforward explanation is that accurate classification is primarily driven by higher layers, so one actually doesn’t need most of the dimensions in the hash code for good classification performance (similar “cache” models using high layer features have been proposed before: e.g. Dubey et al., CVPR 2019; Orhan, NeurIPS 2018; Khandelwal et al., ICLR 2020).

(7) Most importantly, one of the main phenomena observed in this paper (the change of redundancy ratio over training) has already been reported in Hanin and Rolnick (ICML, 2019): they note that the number of linear regions in a deep ReLU net first decreases and then increases during training (please read their section 3 carefully). This would easily explain the trajectory of the redundancy ratio observed in Figure 3c (and in Figure 4a) in this paper. Moreover, the concept of diameter is also rigorously defined (and diameters of linear regions studied) in Hanin and Rolnick (ICML, 2019), but this is not acknowledged at all by the authors. This is a pretty serious omission.

(8) Typos: should be: “Mapping induced by a relu network” (p. 1), “it is worth noting” (p. 2)
“Geometric properties of linear regions” (p.3). “Another diameters of linear regions”? (p. 5)

---

> ### Author Response · Authors · 2020-11-14
> **To Reviewer #1 (3/3)**
>
> Q6.2: _Moreover, no effort is made by the authors to understand what drives good test accuracy in these experiments. A very straightforward explanation is that accurate classification is primarily driven by higher layers, so one actually doesn’t need most of the dimensions in the hash code for good classification performance (similar “cache” models using high layer features have been proposed before: e.g. Grave et al., ICLR 2017; Orhan, NeurIPS 2018; Khandelwal et al., ICLR 2020)._
>
> A6.2: Thanks. We have added a layer-wise ablation study for understanding the behaviours of the neural code of some layers. The results show that the earlier layers have fairly low redundancy ratios but relatively poor categorization accuracies, while higher layers have very poor redundancy ratios though the categorization accuracies are fairly good. One can only observe both encoding properties are satisfied in neural code formed by all layers. Meanwhile, the two properties collectively measure the expressive power of a neural network. Thus, our results suggest that the neural code formed by any part of the neural network is not sufficient compared with that formed by the whole neural network (i.e., all layers).
>
> Q7.1: _Most importantly, one of the main phenomena observed in this paper (the change of redundancy ratio over training) has already been reported in Hanin and Rolnick (ICML, 2019): they note that the number of linear regions in a deep ReLU net first decreases and then increases during training (please read their section 3 carefully). This would easily explain the trajectory of the redundancy ratio observed in Figure 3c (and in Figure 4a) in this paper._
>
> A7.1: Thanks. We respectfully argue that our results are different. We focus on whether almost every linear region has no more than one datum. By contrast, Hanin and Rolnick (ICML, 2019) study the number of linear regions (linear region counting). They are two significantly different facets of the linear partition in the input space. Moreover, we discover that the encoding properties do not apply beyond the training data distribution, no matter how the linear region counting change; please refer to Figure 3. This suggests that the increase of linear region counting would just happen in a small part of the input space which is usually extremely large. Therefore, an increasing linear region counting cannot guarantee a decreasing redundancy ratio.
>
> Q7.2: _Moreover, the concept of diameter is also rigorously defined (and diameters of linear regions studied) in Hanin and Rolnick (ICML, 2019), but this is not acknowledged at all by the authors. This is a pretty serious omission._
>
> A7.2: Thanks. We have carefully discussed this and acknowledge their contributions in the updated manuscript. However, we respectfully argue that our definition is different from theirs, and has opposite behaviours in the context of linear regions (which are extremely irregular).
>
> “The definition in Hanin and Rolnick (ICML, 2019) is ‘the typical distance from a random input to the boundary of its linear region.’ Meanwhile, our diameter is intuitively the longest distance between two points in a linear region. When the linear region is an ideal ball, their distance is equal to or smaller than the radius of the ball, the half of our diameter. However, linear regions are usually extremely irregular in practice. Please refer to a visualization of the linear regions in Figure 1, Hanin and Rolnick (ICML, 2019). Given this, the distances of Hanin and Rolnick (ICML, 2019) would be significantly smaller than our diameter. Overall, these two definitions would exhibit a significant discrepancy depending on the irregular level; one can be even fixed when the other is significantly changed. Moreover, their distance can yield a lower bound for the linear region volume, while ours can deliver an upper bound.”
>
> Q8: _Typos: should be: “Mapping induced by a relu network” (p. 1), “it is worth noting” (p. 2) “Geometric properties of linear regions” (p.3). “Another diameters of linear regions”? (p. 5)_
>
> A8: Thank you and addressed.
>
> [1] Jaeger H. The “echo state” approach to analysing and training recurrent neural networks. GMD Technical Report 148. German National Research Center for Information Technology. 2001.
> [2] Maass W, Natschläger T, Markram H. Real-time computing without stable states: A new framework for neural computation based on perturbations. Neural computation. 2002.

---

> ### Author Response · Authors · 2020-11-14
> **To Reviewer #1 (2/3)**
>
> Q3: _This brings me to the related works section. This section is written very shallowly, the authors do not do a good job of situating their contributions in the context of prior works. You cannot just say: “Other advances in linear region counting include …” (p. 3) and then cite a bunch of references. You have to tell us what each of these papers did and how what you’re doing in this paper differs from these earlier works and makes a meaningful and novel contribution to the prior literature._
>
> A3: Thanks. We provide a detailed review below which will be added to the updated version below. To our best knowledge, this is the first work presents and studies the encoding properties exhibited in the activation patterns.
>
> “Hu & Zhang (2018) and Zhu et al. (2020) prove upper bounds for the linear regions counting in neural networks with piecewise linear activations. Xiong et al. (2020) study the linear region counting of convolutional neural networks. Raghu et al. (2017) define a trajectory length based on activation patterns for measuring the expressiveness of neural networks. Serra et al. (2018) prove lower bounds and upper bounds for the linear region counting which show that one can obtain larger linear region counts when the layer monotonously decreasing from the early layer to the final layer. Kumar et al. (2019) empirically demonstrate that a large proportion of the ReLU activations are always either activated or de-activated for all training examples in a well-trained and fixed network.”
>
> Q4: _The concept of redundancy ratio seems to play a central role in the paper, but it is not defined formally, just a verbal (potentially ambiguous) definition is given. Please define this concept formally to avoid any ambiguities. I’m also not sure this concept is the right one for quantifying the goodness of a code: consider a case where each point in a dataset shares its linear region with exactly one other point in the dataset vs. a case where all points belong to the same linear region. It seems that the redundancy ratio will be 100% in both cases, however intuitively the code in the first case should be much better than in the second case (for example for retrieval)._
>
> A4: Thanks. The formal definition is given as follows.
>
> Definition 1 (redundancy retio): Suppose there are $n$ examples in a dataset $S$. If they are located in $m$ activation regions, the redundancy ratio is defined to be $\frac{n - m}{n}$.
>
> Our experiments are conducted according to this definition. The two examples you raised can be clearly differentiated: the redundancy ratios are 50% and 100%, respectively. Please accept our apologies for any ambiguity. We have carefully updated all related parts of our manuscript.
>
> Q5: _Relatedly, Figure 3c suggests that the redundancy ratio is close to zero for an untrained random network. Then, by the authors’ own definition, the encoding is actually very good before training. Please consider what this means (also see point 7 below)._
>
> A5: Thanks. We respectfully note that the redundancy ratio is only one facet of the encoding properties. The categorization accuracy is also an important part in measuring the encoding properties. However, the categorization accuracy is quite poor in untrained networks. It suggests that the encoding properties are not good before training. Moreover, this phenomenon coincides with the “reservoir effects” [1-2] as Reviewer #3 suggested. We also give an explanation below. It has been added in the manuscript.
>
> “When the neural network is randomly initialized, the input space is randomly partitioned into multiple activation regions. If these activation regions are sufficiently small, almost every training datum has its own activation region. However, the mapping from input data to the output prediction is meanless at random initialization; the neural network may give two completely different predictions to two datums from neighboring activation regions. Therefore, the categorization accuracy is poor at this time.”
>
> Q6.1: _Unfortunately, I don’t think classification results for MNIST only are very meaningful. Almost anything will get above 99% accuracy on MNIST._
>
> A6.1: Thanks. We have conducted experiments on CIFAR-10. The results fully support our arguments. Please refer to our post to all reviewers and the updated manuscript.

---

> ### Author Response · Authors · 2020-11-14
> **To Reviewer #1 (1/3)**
>
> Thank you for your thorough review and detailed comments. All your concerns have been carefully responded. We have updated our manuscript accordingly. Specifically, we would love to clarify our motivation, the dimension of neural code (it can be significantly smaller than the data dimension), and the relationships between our results and existing results. We also present additional experimental results accordingly to your advice. We sincerely hope our responses can fully address your concerns and the merit of this paper can be reevaluated. Much appreciated.
>
> Q1.1: _First of all, I am a bit confused about the motivation behind this work. The motivation is initially couched in terms of hash codes, so one gets the impression that the authors are going to propose a new hash coding scheme using neural networks. But this is clearly not the case._
>
> A1.1: Thanks. Our paper studies the fundamental mechanism of deep learning. The linear partition of the input space would paly a major role in explaining the expressive power of deep learning. To date, the understanding of this linear partition is still premature. Significant efforts are really necessary. Detailed justifications for the motivation and significance of our study are listed below:
>
> 1. To our best knowledge, this is the first work that presents and studies the encoding properties exhibited in the activation patterns. We find in many popular deep learning models: (1) almost every linear region contains at most one training example; and (2) according to the neural code, simple algorithms can achieve fairly good performance on both training and test data. These two properties collectively serve as an informative measure for the expressive power of a neural network.
> 2. Our systematic experiments show that model capacity, training time, sample size, and regularization influence the above two encoding properties. Especially, the first three factors have greater impacts than regularization. These findings shed lights to the understanding of the mechanism under the expressive power of a neural network.
> 3. The neural code may have significant potentials in inspiring new algorithms. As stated in A1.2, a neural code with the encoding properties can have a significantly lower dimensionality than the data ($40$ vs. $784$). It can be used as a hash code. Another straightforward future research direction is to deploy the neural code to help classification.
>
> Q1.2: _The proposed coding scheme is practically useless as a hash code because of the enormous dimensionality of the codes (I estimate this to be on the order of millions even for the toy MNIST case studied in the paper, but it would actually be very helpful for the reader if the authors explicitly mentioned the dimensionality of the proposed hash codes--at least the order of magnitude--)._
>
> A1.2: Thanks. We respectfully argue that the dimension of a neural code having the above two properties can be significantly lower than the data dimension and dataset size. It can be used as a hash code.
>
> For example, we present results of one-hidden-layer MLPs of width from $5$ to $100$ in the experiments on the MNIST dataset. The dimension of neural code equals to the width, from $5$ to $100$. Our experiments show that the logistic regression accuracy on the neural code has been higher than $0.9$,  when the dimension of neural code is $40$. In contrast, the data dimension is $784$, the training sample size is $60,000$, and the test sample size is $10,000$. Please refer to Figure 2(b), Sections 5.2, 5.3, 5.4, and Appendix A.1 for more details.
>
> Q1.3: _This is also why the authors are stuck with the toy MNIST dataset throughout the paper, because the proposed scheme is completely impractical for any reasonably large dataset and model._
>
> A1.3: Please refer to A1.1 and A1.2. Thank you.
>
> Q2: _This begs the question: what exactly is the significance of the observation that linear regions satisfy some properties of hash codes, if they’re not going to be used as hash codes? It’s not meaningful to just point out that something satisfies some properties of hash codes. One can point to a million different things that satisfy some properties of hash codes. What exactly is the significance of linear regions having these properties?_
>
> A2: Please refer to A1.1 and A1.2. Thank you.

---

### Official Review · AnonReviewer2 · 2020-10-28
**MNIST alone is not enough to draw any decisive conclusions**

**Rating:** 5
**Confidence:** 4

**Review:**

This draft proposes to use the relu activation pattern of the neurons in the neural network as the hash code for the input. Essentially the input features are bucketized into small piecewise linear regions. The authors show empirically that the proposed hash code has small collision and high accuracy given certain conditions including
1. The features are around the sample manifold
2. the training time is long enough
3. the network is wide enough
4. the training sample size is large enough
The authors also found empirically the effect of regularization is relatively small on the encoding properties.

I feel this is an interesting thought but I am not sure if this is the first work on it. Some other possible issues:
1. Figure 4(a) somehow shows the redundancy is the smallest at epoch 0, then it goes high after 1 epoch and decreases slowly as the number epochs grows. Can authors provide some explanation on this? Does it suggest the random initialization of the neural network gives a good hash code in terms of the redundancy metric? (of course the accuracy will be bad)
2. The authors used K-means as another benchmark to compare. To me k-means is an unsupervised clustering algorithm. How do you get the accuracy from k-means? How do you match the cluster id to the labels?
3. My biggest complaint is that only the MNIST data set is investigated in the experiment. MNIST is too easy to show any conclusive results. You may need to work on other data sets such as image net, cifar 100, or NLP related data set to draw a convincing conclusion.
4. All the conclusions are purely empirical. Can authors provide some explanation or intuition on why the redundancy ratio decreases as the training time grows? Is this related to the type of optimizer being used? Why can a larger sample size also help reduce the redundancy ratio?


Overall I think this draft has some really good ideas but the empirical result is not quite conclusive due to the lack of extensive experimentation.

---

> ### Author Response · Authors · 2020-11-14
> **To Reviewer #2**
>
> Thank you very much for your thorough review and constructive feedback. All your concerns have been carefully responded. We have updated our manuscript accordingly.
>
> Q0: _I feel this is an interesting thought but I am not sure if this is the first work on it._
>
> A0: Thanks for recognizing our work. To our best knowledge, this is the first work that presents and studies the encoding properties exhibited in the activation patterns.
>
> Q1: _Figure 4(a) somehow shows the redundancy is the smallest at epoch 0, then it goes high after 1 epoch and decreases slowly as the number epochs grows. Can authors provide some explanation on this? Does it suggest the random initialization of the neural network gives a good hash code in terms of the redundancy metric? (of course the accuracy will be bad)._
>
> A1: Thanks. When the neural network is randomly initialized, the input space is randomly partitioned into multiple activation regions. If these activation regions are sufficiently small, almost every training datum has its own activation region. However, the mapping from input data to the output prediction is meanless at random initialization; the neural network may give two completely different predictions to two datums from neighboring activation regions. Therefore, the categorization accuracy is poor. This phenomenon coincides with the “reservoir effects” [1-2] as suggested by Reviewer #3. It also suggests that just redundancy ratio is not sufficient to measure the encoding properties. We have carefully discussed this in our manuscript.
>
> Q2: _The authors used K-means as another benchmark to compare. To me k-means is an unsupervised clustering algorithm. How do you get the accuracy from k-means? How do you match the cluster id to the labels?_
>
> A2: Thanks. The pipeline for the experiments on $K$-means is as follows: (1) we set $K$ as the number of classes; (2) run $K$-means on the neural codes and obtain $K$ clusters; (3) every cluster can be assigned a label from $\{1, 2, \ldots, 10\}$. Thus, there are $90$ (cluster, label) pairs; (4) for every (cluster, label) pair, we assign the label to all datums from the cluster and calculate the accuracy; and (5) we select the highest accuracy as the accuracy of the $K$-means algorithm. We have carefully discussed this in our manuscript.
>
> Q3: _My biggest complaint is that only the MNIST data set is investigated in the experiment. MNIST is too easy to show any conclusive results._
>
> A3: Thanks. We have conducted experiments on the CIFAR-10 dataset. The results still support our findings. Please refer to our post to all reviewers and the updated manuscript.
>
> Q4.1: _All the conclusions are purely empirical. Can authors provide some explanation or intuition on why the redundancy ratio decreases as the training time grows?_
>
> A4.1: Thanks.  Our hypothesis is that the diameters of activation regions around training examples decreases while the training progressing. Please also refer to Hypothesis 1 in the manuscript. Intuitively, this reflects that the neural network is fitting the training data. Moreover, our experiments show that the encoding properties do not apply beyond the training data. In other words, the redundancy ratio on data beyond the training data distribution would not decrease as the training progresses. This suggests that the neural network training is focused around the training data.
>
> Q4.2: _Is this related to the type of optimizer being used?_
>
> A4.2: Thanks. We do not observe any significant discrepancy of the encoding properties caused by the optimizers SGD and Adam on both MNIST and CIFAR-10.
>
> Q4.3: _Why can a larger sample size also help reduce the redundancy ratio?_
>
> A4.3: Thanks. Intuitively, a larger training sample size supports the neural network to attain a higher expressive power, i.e., the linear partition in the input space is finer. Meanwhile, a sample of larger size requires a finer linear partition to yield the same redundancy ratio. Our experiments show that the first effect is stronger than the second one. Thus, a larger sample size can help reduce the redundancy ratio.
>
> Q5: _Overall I think this draft has some really good ideas but the empirical result is not quite conclusive due to the lack of extensive experimentation._
>
> A5: Thank you very much for recognizing our contribution. We have presented more empirical results on the CIFAR-10 dataset, layer-wise ablation study, random data, and random labels. Please refer to our post to all reviewers and the updated manuscript.
>
> [1] Jaeger H. The “echo state” approach to analysing and training recurrent neural networks. GMD Technical Report 148. German National Research Center for Information Technology. 2001.
> [2] Maass W, Natschläger T, Markram H. Real-time computing without stable states: A new framework for neural computation based on perturbations. Neural computation. 2002.

---

### Author Response · Authors · 2020-11-14
**To all reviewers: More experimental results presented; code submitted; concerns responded**

The authors sincerely appreciate all the four reviewers for the thorough review and constructive feedback. Following the reviewers’ advice, we will present here the empirical results regarding the CIFAR-10 dataset, layer-wise ablation study, random data, and random labels. We have also submitted our source code package as supplementary material to secure the reproducibility. Our code, trained models, and collected data will be released publicly. For other comments, we will respond to the specific reviewer.

In this post, we overview the new empirical results. We have also submitted figures as supplementary material. Please refer to the updated manuscript.

---

**1. Experiments of MLPs on CIFAR-10**

We first trained five-hidden-layer MLPs on CIFAR-10. The plots are given in the "Additional empirical results.pdf" in the supplementary material.

Our experiments show that (1) the redundancy ratio can be almost $0$ on CIFAR-10; (2) the categorization accuracy on neural code is almost the same the test accuracy of the corresponding trained model on the raw data; and (3) the two encoding properties have significant correlations with model size, training time, and training sample size.

---

**2. Results for CNNs on CIFAR-10**

We then trained VGG-19, ResNet-18, ResNet-20, and ResNet-32 on CIFAR-10. We applied logistic regression to verify the categorization accuracy due to time limitation.

Our experiments show that (1) the redundancy ratio is always $0$; and (2) the categorization accuracy on neural code is fairly high as the following table. These results suggest the encoding properties hold in the CNNs trained on CIFAR-10.

|Architecture|VGG-19|ResNet-18|ResNet-20|ResNet-32|
|---|:-:|:-:|:-:|:-:|
|LR acc (/%)|92.19|89.55|88.76|89.05|
|Test acc* (/%)|91.43|90.42|90.44|90.45|

*Test acc refers to the test accuracy of the corresponding models on the raw data.

---

**3. Layer-wise ablation study**

The implementations are the same as Sections 1 and 2.

Our experiments show that (1) the earlier layers have fairly low redundancy ratios but relatively poor categorization accuracies; (2) higher layers have relatively poor redundancy ratios though the categorization accuracies are fairly good (approximately equal to the categorization accuracies of the neural code formed by all layers); (3) from the first/last layer towards the full network, the two encoding properties gradually change; (4) the correlations between the encoding properties and model size, training time, and training sample size still hold in the neural code of one single layer; and (5) one can hardly observe both encoding properties are satisfied in the neural code formed by a part of the neural network. The second property also suggests that one can use the categorization accuracies of final layers to estimate the categorization accuracies of the neural code formed by all layers.

|Layer|1|2|3|4|5|
|---|:-:|:-:|:-:|:-:|:-:|
|R ratio (/%)| 0.05|0.43|3.95|9.47|26.57|
|$K$-Means (/%)|23.12 | 29.67 | 41.00 | 44.87 | 57.86 |
|$K$-NN (/%)| 34.04 | 42.15 | 44.37 | 46.40 | 48.42 |
|LR (/%)| 33.75 | 42.79 | 46.42 | 47.64 | 49.80 |

|Layers|1|1-2|1-2-3|1-2-3-4|1-2-3-4-5|
|---|:-:|:-:|:-:|:-:|:-:|
|R ratio (/%)|0.05|0.01|0|0|0|
|$K$-Means (/%)|23.12|28.28|34.83|44.85|51.05|
|$K$-NN (/%)|34.04|44.42|48.92|50.15|50.41|
|LR acc (/%)|33.75|44.50|48.43|49.22|50.15|

|Layers|5|4-5|3-4-5|2-3-4-5|1-2-3-4-5|
|---|:-:|:-:|:-:|:-:|:-:|
|R ratio (/%)|26.57|0.32|0.01|0|0|
|$K$-Means (/%)|57.86|56.94|54.32|50.96|51.05|
|$K$-NN (/%)|48.42|49.01|49.29|50.07|50.41|
|LR acc (/%)|49.80|50.43|50.49|49.96|50.15|

The results for the correlations are given in the "Additional empirical results.pdf" in the supplementary material.

---

**4. Experiments on random data**

We generated $60,000$ images of size $28 \times 28$ whose pixels are drawn from uniform distribution $U(0,1)$. We tried to train many MLPs and CNNs on the generated data. The training fails to converge in every case. Some plots are given in the "Additional empirical results.pdf" in the supplementary material.

---

**5. Experiments on noisy labels**

We introduce label noise into MNIST with different noise rates. Then, multiple one-hidden-layer MLPs are trained on the generated data. Our experiments show that the encoding properties still stand though become relatively worse. It suggests that the structure of the input-data can drive the organization of the hashed space. Plots are given in the "Additional empirical results.pdf" in the supplementary material.

---

> ### Author Response · Authors · 2020-11-17
> **Manuscript updated**
>
> Following the reviewers’ advice, we have carefully updated our manuscript. We have also submitted a draft in the supplementary material, where all revisions are highlighted in blue. We sincerely hope all concerns have been addressed. Thank you very much for the constructive comments!

---

### Decision · Program_Chairs · 2021-01-07
**Final Decision**

**Decision:**

Reject

**Comment:**

The paper studies the behavior of the intermediate ReLU-like activations of trained neural networks and show empirically that the intermediate activation can be used as a hashing function for the examples with some key advantages, including almost no collisions and that there are desirable geometric properties (i.e. can use k-means, k-nn, logistic regression on these embeddings).

Pros:
- The experimental analysis was solid and thorough investigating the effects of model size, training time, training set, regularization and label noise.

Cons:
- An overall lack of novelty. It is already quite well-known throughout the ML community that in many cases, using the intermediate embeddings serve as useful features to apply more classical methods such as kNN and clustering.

Overall, the reviewers appreciated the solid and thorough investigation into the hashing properties of neural network activation patterns, which convincingly confirms some intuitions about the behavior of activation patterns in neural networks. However, the reviewers also agreed that there was no significant new finding. There have already been many studies on clustering and kNN on the embeddings of a network.
Thus, the core novelty of the paper appears to be the finding that almost every linear region has at most one datapoint after training (which does not seem too surprising given Hanin & Rolnick (ICML 2019)); however, without further novel implications of this finding, the impact of the paper is limited.